# TIA1 Loss Exacerbates Fatty Liver Disease but Exerts a Dual Role in Hepatocarcinogenesis

**DOI:** 10.3390/cancers14071704

**Published:** 2022-03-27

**Authors:** Dobrochna Dolicka, Szabolcs Zahoran, Marta Correia de Sousa, Monika Gjorgjieva, Christine Sempoux, Margot Fournier, Christine Maeder, Martine A. Collart, Michelangelo Foti, Cyril Sobolewski

**Affiliations:** 1Department of Cell Physiology and Metabolism, Translational Research Centre in Onco-Hematology (CRTOH), Faculty of Medicine, University of Geneva, CH-1211 Geneva, Switzerland; dobrochna.dolicka@unige.ch (D.D.); marta.sousa@unige.ch (M.C.d.S.); monika.gjorgjieva@unige.ch (M.G.); margot.fournier@unige.ch (M.F.); christine.maeder@unige.ch (C.M.); michelangelo.foti@unige.ch (M.F.); 2Department of Microbiology and Molecular Medicine, Faculty of Medicine, University of Geneva, CH-1211 Geneva, Switzerland; szabolcs.zahoran@unige.ch (S.Z.); martine.collart@unige.ch (M.A.C.); 3Service of Clinical Pathology, Institute of Pathology, Lausanne University Hospital and University of Lausanne, CH-1007 Lausanne, Switzerland; christine.sempoux@chuv.ch

**Keywords:** TIA1, stress granules, NASH, HCC, oncogenes, tumor suppressors

## Abstract

**Simple Summary:**

Hepatocellular carcinoma (HCC) is a leading cause of cancer mortality worldwide with few and poorly efficient therapeutic options. Adenylate–uridylate-rich-element-binding proteins (AUBPs) are potent post-transcriptional regulators of gene expression, whose alterations have been associated with the development of chronic liver diseases (e.g., fatty liver disease) and their progression toward HCC. Emerging evidence indicates that T-Cell-restricted intracellular antigen-1 (TIA1), an AUBP regulating mRNA translation, promotes HCC. Herein, we further characterize TIA1’s functions in hepatic cancer cells by identifying an entire set of oncogenes and tumor suppressors under TIA1’s influence, thus suggesting a dual role in hepatocarcinogenesis. In agreement, using in vivo models, we found that contrary to our observations in HCC cells, TIA1 can exert a tumor suppressive function and refrain from hepatic steatosis and fibrosis. Finally, TIA1 expression is reduced in human HCC. Our study provides new insights regarding the role for TIA1 in HCC and questions the current concept of TIA1 as a strict tumor promoter.

**Abstract:**

Alterations in specific RNA-binding protein expression/activity importantly contribute to the development of fatty liver disease (FLD) and hepatocellular carcinoma (HCC). In particular, adenylate–uridylate-rich element binding proteins (AUBPs) were reported to control the post-transcriptional regulation of genes involved in both metabolic and cancerous processes. Herein, we investigated the pathophysiological functions of the AUBP, T-cell-restricted intracellular antigen-1 (TIA1) in the development of FLD and HCC. Analysis of TIA1 expression in mouse and human models of FLD and HCC indicated that TIA1 is downregulated in human HCC. In vivo silencing of TIA1 using AAV8-delivered shRNAs in mice worsens hepatic steatosis and fibrosis induced by a methionine and choline-deficient diet and increases the hepatic tumor burden in liver-specific PTEN knockout (LPTENKO) mice. In contrast, our in vitro data indicated that TIA1 expression promoted proliferation and migration in HCC cell lines, thus suggesting a dual and context-dependent role for TIA1 in tumor initiation versus progression. Consistent with a dual function of TIA1 in tumorigenesis, translatome analysis revealed that TIA1 appears to control the expression of both pro- and anti-tumorigenic factors in hepatic cancer cells. This duality of TIA1′s function in hepatocarcinogenesis calls for cautiousness when considering TIA1 as a therapeutic target or biomarker in HCC.

## 1. Introduction

Hepatocellular carcinoma (HCC) represents the seventh most common cancer and the second cause of cancer mortality worldwide [1]. HCC can arise as an end stage of chronic liver disorders, including viral infections, alcohol consumption and non-alcoholic fatty liver disease (NAFLD) [1]. NAFLD develops from an abnormal accumulation of lipids in the liver (steatosis) and can progress to non-alcoholic steatohepatitis (NASH), fibrosis and cirrhosis [2]. Cirrhosis represents a cause of morbidity and a major risk factor for tumor development. Of note, HCC can also develop directly from NASH through poorly characterized mechanisms. However, the most frequent mutations (e.g., *TP53, CTNNB1*) leading to HCC development are currently poorly druggable and do not explain the development of HCC in a non-cirrhotic context [3]. Our recent work has uncovered a whole network of oncogenes (ONC) and tumor suppressors (TS) promoting NASH and HCC (e.g., S100A11), thus pointing to the importance of mutation-independent alterations in the development of HCC [4]. In agreement, increasing evidence indicates that epigenetic and post-transcriptional alterations driving the overexpression of ONCs and the loss of TS represent a major opportunity to identify new and efficient biomarkers and therapeutic targets for cancers with poorly efficient therapeutic options.

Deregulated gene expression at the post-transcriptional level importantly contributes to the development of a wide range of diseases, including metabolic, inflammatory diseases and cancers. In this regard, a deregulation of specific RNA-binding proteins, i.e., adenylate–uridylate-rich-elements binding proteins (AUBPs), has been suggested to strongly impact NAFLD, NASH and hepatic cancers development [5,6]. Following their binding to adenylate–uridylate-rich elements (ARE) within the 3′-untranslated regions (3′-UTRs) of mRNAs, these proteins can regulate mRNA degradation and/or translation, thus potentially leading to abnormal overexpression of ONCs, inflammatory factors or contrarily to TSs silencing [7]. Therefore, deregulated expression or activity of AUBPs can induce similar outcomes as genetic mutations. HCC development is tightly associated with the deregulated activity of various AUBPs, e.g., HuR [8] or T-cell-restricted intracellular antigen-1 (TIA1) [9]. Moreover, our previous work has uncovered other AUBPs deregulated in human HCC [10]. This study identifies TIA1 as a potential TS, whose function in HCC remains poorly known.

Together with various co-factors (e.g., TIAR), TIA1 acts as a translational repressor during stress conditions, where it sequesters its target mRNAs into small cytoplasmic foci known as “stress granules” (SGs). A canonical model states that there, mRNAs are kept translationally silenced and are sorted to re-enter translation or proceed to mRNA decay, depending on the model and stress conditions [11]. SGs can sequester pro-apoptotic factors, such as RACK1 or TRAF2 [12], therefore representing an important survival mechanism for cancer cells. Depending on the cancer type, TIA1 was reported to behave either as a TS or as an ONC. Indeed, in colorectal cancer, TIA1 downregulation contributes to the induction of pro-inflammatory mediators (e.g., COX-2), thus promoting cancer growth [13,14], while in esophageal squamous cell carcinoma, TIA1 acts as an oncogene by protecting the mRNAs from other oncogenes (e.g., *SKP2*, *CCNA2*) [15]. More recently, an oncogenic function of TIA1 was suggested in a hepatic cancer cell line (HepG2) based on its capacity to silence the TS gene *IGFBP3* [9]. In addition, SGs were recently reported to be associated with resistance to sorafenib, a drug commonly used in HCC therapy, in hepatic cancer cell lines [16]. Together, these previous in vitro data suggest that TIA1 and SG formation might be involved in HCC development. However, the role, function and regulation of TIA1 as well as the biogenesis of SGs in liver carcinogenesis remain poorly understood.

Herein, we investigated in vivo the role for TIA1 in hepatic steatosis, inflammation, fibrosis and HCC development. We also identified TIA1′s targets potentially implicated in its pathophysiological role and assessed its role in SG formation in hepatic cancer cells.

## 2. Materials and Methods

### 2.1. Animals

#### 2.1.1. Animal Housing

Mice were kept at 23 °C in a 12 h cycle, in cages containing enrichment (disposable house and cotton cocoons) with access to food (SAFE-150 diet, SAFE, Augy, France) and water ad libitum. Experiments were performed on males; prior to decapitation, mice were anaesthetized with isoflurane (ISO250, Rothacher-Medical, Heitenried, Switzerland). Experiments were performed in accordance with standards of the “Animal Research Reporting of In Vivo Experiments” (ARRIVE, https://www.nc3rs.org.uk/arrive-guidelines, accessed on 1 December 2021). All experiments were ethically approved by the Geneva Health head office and were conducted in agreement with the Swiss guidelines for animal experimentation (authorization GE/120/20).

Male *db/db* and corresponding control mice were purchased from Charles River Laboratories (Zurich, Switzerland) (BKS.Cg-Dock7m+/+LeprdbJ). Liver samples from male ob/ob and control mice (B6.V-Lepob/JRj) were provided by Françoise Rohner-Jeanrenaud (University of Geneva, Geneva, Switzerland). 

#### 2.1.2. AAV8 Injection

Mice were injected with adeno-associated virus serotype 8 (AAV8) vectors containing an shRNA for TIA1 (AAV8-GFP-U6-m-TIA1-shRNA, Vector Biolabs, Malvern, PA, USA, shTIA1) or a scrambled control (AAV8-GFP-U6-scrmb-shRNA, Vector Biolabs, shCtl). C57BL6/J mice were injected at 2 months of age with a dose of 2 × 10^11^ viral particles and LPTENKO mice were injected at 8.5 months with a dose of 2 × 10^11^ viral particles, followed by a second dose (1 × 10^11^ viral particles) at 10.5 months of age.

#### 2.1.3. Computed Tomography (CT)

Animals were imaged by CT using Quantum GX microCT software (PerkinElmer, Waltham, MA, USA). Mice were imaged at 8.5 and 11.5-months. Before the first scan, a single dose of 100 μL ExiTron nano 6000 (Viscover, Berlin, Germany) was injected retro-orbitally. CT scan analysis was performed using OsiriX MD v.10.0.1 software.

#### 2.1.4. Methionine and Choline Deficient (MCD) Diet

2.5-months-old mice were fed with an MCD (E15653-94, ssniff, Soest, Germany) or Chow (E15654-04, ssniff) diet for 12 days.

#### 2.1.5. Serum Analysis

Blood retrieved during sacrifices was centrifuged at 1500× *g* for 10 min to obtain plasma. Serum was analyzed using the cobas 8000 system (Roche, Basel, Switzerland). Low-density lipoprotein (LDL) concentration was calculated as a difference between total cholesterol and high-density lipoproteins (HDL).

### 2.2. In Vitro Cultures

Huh7 cells (JCRB0403) were purchased from the Japanese Collection of Research Bioresources Cell Bank and Sekisui Xenotech (Cambridge Cir Dr, KS, USA). HepG2 human hepatoma cell line was purchased from ATCC (Manassas, VA, USA). HepaRG cells were previously generated by Christiane Guillouzo, Philippe Gripon and Christian Trepo [17,18] and provided by Biopredic International (Saint-Grégoire, France). Hepa1-6 cells were obtained from Pr. Manlio Vinciguerra (Institute for Liver and Digestive Health, University College London, UK). SNU398 and Hep3B were provided by Caroline Gest (Inserm U1053, University of Bordeaux, France). THLE-2 cells were provided by Britta Skawran (Medizinische Hochschule Hannover, Institut fur Humangenetik, Hannover). THLE-2 were cultured in BEGM media (BEGM Bullet Kit, Lonza, Basel, Switzerland), supplemented with 10% fetal bovine serum (FBS, Thermo Fisher Scientific, Waltham, MA, USA), 70 ng/mL phosphoethanolamine (Biochrom, Cambridge, UK) and 5 ng/mL EGF (BEGM Bullet Kit, Lonza, Basel). Huh7, HepG2, Hepa1-6 and Hep3B cells were cultured in DMEM medium (glucose 1 g/L, Thermo Fisher Scientific) supplemented with 10% FBS (Thermo Fisher Scientific) and 1% penicillin–streptomycin (PS, Thermo Fisher Scientific). AML12 cells were cultured in DMEM F12 medium supplemented with 100 nM of dexamethasone (reconstituted in ethanol, Sigma–Aldrich), 10% FBS 1% PS and 1% of insulin-transferrin-selenium (ITS-G, Thermo Fisher Scientific). HepaRG cells were cultured in William’s medium (Gibco, Waltham, MA, USA) supplemented with 10% FCS, 10^−9^ M insulin and 10^−6^ M hydrocortisone (Sigma–Aldrich, Saint-Louis, MO, USA). For differentiation, HepaRG cells were maintained at full confluence for 14 days in normal medium, followed by medium supplemented with 2% of dexamethasone (DMSO) for additional 14 days. SNU398 were cultured in RPMI GlutaMAX medium (Thermo Fisher Scientific) supplemented with 10% FCS and 1% PS.

#### 2.2.1. siRNA Transfection

Cells were seeded at a density of 20,000 cells/cm^2^ and transfected on the following day with 20 nM of target siRNA or AllStars negative control siRNA (siTIA1 or siCtl, respectively, Qiagen, Basel, Switzerland) using 0.6 μL INTERFERIN (Polyplus transfection, Illkirch, France) per 10 cm^2^ according to manufacturer’s protocol. Further analyses were performed 48 h post-transfection. siRNA sequences are provided at the end of Appendix A.

#### 2.2.2. Plasmid Transfection

Cells were seeded at a density of 40,000 cells/cm^2^ and transfected on the following day with 0.5 μg of plasmid using Lipofectamine 3000 (Thermo Fisher Scientific) per 10 cm^2^ according to manufacturer’s protocol. Further analyses were performed 48 h post-transfection. pFRT_TO_eGFP_TIA1 (TIA1-EGFP) was a gift from Thomas Tuschl (Addgene plasmid # 106094; http://n2t.net/addgene:106094; accessed on 1 December 2021; RRID: Addgene_106094) [19]. The pEGFP plasmid (pEGFP-C2, ref#: 6083-1) was acquired from Clontech (now: Takara Bio, San Jose, CA, USA).

#### 2.2.3. Treatments

For anti-cancer drugs’ treatment, cells were seeded at a density of 20,000 cells/cm^2^ and treated with 5 μM (for cell proliferation assays) or 20 μM (for immunofluorescence assays) of sorafenib for 2 or 24 h, respectively. In case of cells with TIA1 knock-down, cells were transfected as described in “siRNA transfection” (Section 2.2.1) and treated with the drug 48 h later.

#### 2.2.4. Primary Hepatocytes Isolation

Mouse primary hepatocytes (MPH) were isolated as previously described [20]. Briefly, mouse livers were perfused through the portal vein with a collagenase-containing solution (C5138, Sigma–Aldrich). Cells were then separated by density gradient centrifugation in Percoll (17-0891-01, GE Healthcare, Danderyd, Sweden). MPH were plated on collagen-coated culture plates in Williams medium supplemented with 10% FBS, 1% PS, 1% GlutaMax (Thermo Fisher Scientific), 1 µM dexamethasone and 5 µg/mL insulin. Human primary hepatocytes (HPH) were kindly provided by Leo Bühler (HUG, Geneva, Switzerland).

### 2.3. Cellular Assays

#### 2.3.1. Cell Proliferation Assay

A number of 20,000 cells/cm^2^ were seeded and transfected on the following day using an siRNA against TIA1 or an siCtl (siRNA sequences are provided at the end of Appendix A). Twenty-four hours post-transfection the cells were reseeded at a density of 20,000 cells/cm^2^ and another 24 h, 48 h and 72 h later the cells were detached using Trypsin-EDTA 0.25% (Thermo Fischer Scientific) and counted using a Neubauer chamber. For cells treated with sorafenib, cells were transfected with siRNA, treated with sorafenib 48 h later and then analyzed after an additional 24 h (due to the reaching of confluence, Huh7 were reseeded at 20,000 cells/cm^2^ 48 h post-transfection).

#### 2.3.2. Cell Cycle Analysis

A number of 20,000 cells/cm^2^ were seeded and transfected on the following day using an siRNA against TIA1 or an siCtl. 24 h later the cells were detached using Trypsin-EDTA 0.25%, counted and fixed with 75% ethanol. Following overnight incubation at 4 °C, the cells were washed with PBS and stained with propidium PI/RNase buffer (BD Biosciences, Franklin Lakes, NJ, USA) according to the manufacturer’s instructions. The cells were then analyzed on an Accuri C6 flow cytometer (BD Biosciences) using FL2A and FL2H channels. Data were analyzed using FlowJo v10 software (BD Biosciences).

#### 2.3.3. TUNEL Assay

The cells were seeded and transfected as described in siRNA transfections. After 72 h, they were detached using Trypsin-EDTA 0.25%, washed with PBS and fixed in 2% paraformaldehyde. The cells then followed another PBS wash and were stored in 70% ethanol for further analysis. On the day of the analysis, the cells were washed and stained as described in the TUNEL FITC Assay Kit (ab66108, Abcam, Cambridge, UK). The cells were then analyzed on CytoFLEX (Beckman Coulter, Brea, CA, USA) and the results were processed using FlowJo (V10, BD Biosciences).

#### 2.3.4. Migration and Invasion Assays

A number of 20,000 cells/cm^2^ were seeded and transfected on the following day using an siRNA against TIA1 or a siCtl. After 24 h, the cells were detached using Trypsin-EDTA 0.25% and reseeded in 48-well Micro Chemotaxis Chambers (Neuro Probe Inc, Gaithersburg, MD, USA). For both types of assays, the lower chambers were filled with DMEM medium (4.5 g/L glucose) supplemented with 10 ng/mL TGFβ (Peprotech, London, UK), while the upper chambers were filled with serum-free DMEM 1 g/L glucose and 50,000 cells/chamber. For invasion assay, membranes were coated with growth factor reduced Matrigel (Corning, Corning, NY, USA). After 24 h, the membranes were fixed in 70% ethanol and stained with hematoxylin for 10 min. Cells that went through the membrane were counted in at least three fields of view using ImageJ software and the Cell Counter plugin.

### 2.4. Real Time PCR

RNA was isolated using Trizol reagent (15596-018, Thermo Fisher Scientific). Reverse transcription was carried out on 0.5–1 μg RNA using the High-Capacity RNA-to-cDNA Kit (4387406, Thermo Fisher Scientific) according to the protocol provided by the manufacturer. Real-time PCR analysis was performed using PowerUp SYBR Green Master Mix (A25778, Thermo Fisher Scientific) on StepOnePlus and QuantStudio systems (Thermo Fisher Scientific). Primer sequences can be found at the end of Appendix A. Cycle threshold values were normalized with housekeeping genes and expressed as ^ΔΔ^Ct compared to control conditions.

### 2.5. Western Blot

Cells/tissues were transferred to RIPA buffer (50 mM Tris-HCl, pH 6.8, 100 mM DTT, 2% SDS, 0.1% bromophenol blue, 10% glycerol). Protein lysates were centrifuged at 12,000× *g* for 10 min, and the supernatant was collected. Protein concentrations were determined using a Bicinchoninic Acid Protein Assay Kit (Pierce Biotechnology, Waltham, MA, USA). Then, 10 µg of protein lysates per sample were added on a 5–20% gradient of sodium dodecyl sulfate–polyacrylamide gel electrophoresis (SDS-PAGE) gels and transferred onto nitrocellulose membranes (RPN303D, Sigma–Aldrich). Membranes were incubated for 1 min at room temperature (RT) in polyvinyl alcohol and kept in primary antibodies solutions overnight at 4 °C. Membranes were then washed with TBS-Tween 0.1% and incubated with secondary antibodies for 1 h at RT. Membranes were incubated with an ECL Prime substrate (RPN22232, Sigma–Aldrich) for 30 s. Signal detection was performed using the PXI/PXI Touch from Syngene (Synoptics group, Cambridge, UK), blots were quantified with ImageJ™ software. For detailed description of antibodies used (see Appendix A). TIA1 antibodies G-3 and C-20 validation blots can be found in Appendix A.

### 2.6. Immunostainings

#### 2.6.1. Preparation of Histological Slides

Tissues were fixed overnight with 4% paraformaldehyde and washed with PBS. Then, the samples were dehydrated and embedded in paraffin. Specimens were then cut into 5 μm sections and stained with hematoxylin and eosin (for morphological analysis) or Masson’s Trichrome/Sirius red (for analysis of fibrosis) and mounted with coverslips. Slides were further analyzed by a pathologist/researcher blinded to the mouse experimental group.

#### 2.6.2. Human Tissue Microarray Immunohistochemistry

Human tissue microarrays (US Biomax, Derwood, MD, USA) were stained against TIA1 using the Abcam IHC–Paraffin protocol. In brief, slides were deparaffinized, rehydrated and heated in citrate buffer for antigen retrieval. Then, they were permeabilized in 0.3% Triton X-100 in TBS for 15 min, blocked with 10% goat serum in BSA for 2 h at RT and incubated overnight at 4 °C with an anti-TIA1 antibody diluted 1:100 in TBS-BSA 3%. On the following day, endogenous peroxidase was blocked by incubation in a 0.3% H2O2 solution for 15 min and then the slides followed an incubation with an anti-goat antibody conjugated with horseradish peroxidase (dilution 1:500) at room temperature for 1 h. Each step was followed by washing in 0.025% Triton X-100 in TBS. The slides were visualized by a 3 min incubation with DAB Substrate Kit (ab64238, Abcam) and counterstained with hematoxylin for 5 min. The results were analyzed using a “–” to “+++” qualitative scale, with “–” signifying no staining and “+++”—intense staining, in a blinded way. Detailed scores are presented in Appendix A. Slides were further analyzed by two blinded researchers. The antibody was validated in HepG2 cells silenced for TIA1 (Appendix A).

#### 2.6.3. Immunocytochemistry

At the end of the experiment, cells were washed with PBS-1X and fixed with 2% paraformaldehyde. Then, they were permeabilized in 0.3% Triton X-100 in TBS for 15 min, blocked with 3% BSA for 1.5 h at RT and incubated overnight at 4 °C with a primary antibody. On the following day, the cells were incubated with a secondary antibody conjugated with horseradish peroxidase at room temperature for 1 h. Each step was followed by washing in 0.025% Triton X-100 in TBS. The slides were visualized by a 3 min incubation with DAB Substrate Kit and counterstained with hematoxylin for 5 min. The specification of antibodies can be found below.

#### 2.6.4. Immunofluorescence

The cells followed a similar protocol to the one described in the Section 2.6.3 until primary antibody incubation. On the following day, the cells were incubated with a fluorescent antibody for 1 h in the dark and then with Hoechst 33342 (1 µg/mL, Sigma–Aldrich) for 10 min. The slides were then mounted with DAKO Fluorescence Mounting Medium (Agilent Technologies, Santa Clara, CA, USA). The samples were imaged using a fluorescent microscope (Evos^®^ FL Cell Imaging System, Thermo Fischer Scientific) and/or Axio Imager.Z2 Basis LSM 800 confocal microscope (Zeiss, San Diego, CA, USA). The images were then processed in ImageJ and Zen Blue (Zeiss), respectively. The specification of antibodies can be found below.

#### 2.6.5. Bodipy Staining

The cells were seeded and transfected as described in siRNA transfection. After 72 h of seeding, the cells were fixed with 2% paraformaldehyde and then stained 30 min at room temperature in the dark with 1 µg/mL BODIPY (boron–dipyrromethene) 493/503 (Molecular Probes, Eugene, OR, USA) and Hoechst 33342 (1 µg/mL). The samples were then imaged using a fluorescent microscope (Evos^®^ FL Cell Imaging System, Thermo Fischer Scientific) and analyzed in ImageJ.

### 2.7. Translatomics Analysis

#### 2.7.1. Polysome Profiling

Polysome profiling experiments were carried out based on different approaches, as previously described in the literature and optimized to our study protocol [21,22]. In brief, 20–60% linear sucrose (Sigma–Aldrich) density gradients were prepared in 40 mM HEPES (pH = 7.5), 40 mM KCl and 20 mM MgCl_2_. In total, five 10 cm plates of HepG2 cells with 70% confluency were treated with 100 µg/mL cycloheximide (CHx) (Sigma–Aldrich) for 30 min on 37 °C. The cells were collected by scraping in ice-cold 1× PBS (supplemented with 100 µg/mL CHx), pelleted by centrifugation (300× *g*, 5 min, 4 °C) and lysed in freshly prepared cell polysome lysis buffer (100 mM KCl; 50 mM TRIS-HCl pH = 7.4; 1.5 mM MgCl_2_; 1 mM DTT; 1 mg/mL heparin; 1.5% NP-40; 100 µg/mL CHx (Sigma-Aldrich), supplemented with EDTA-free protease inhibitor cocktail (Roche) and 100 U/mL SuperaseIn RNase inhibitor (Thermo Fisher Scientific)) for 15 min on ice. The lysate was cleared with centrifugation (15,228× *g*, 4 °C, 20 min) and the absorbance at 260 nm was measured with NanoDrop. The concentrations were equilibrated with the lysis buffer and 400 µL from the supernatants were loaded on the gradients, equal with 160 µg total RNA. The gradients were ultracentrifuged (Optima XPN-100, Beckman) in SW-41Ti swinging bucket rotor (210,000× *g*, 3 h 30 min, 4 °C). The fractions (1 mL for each) were collected with Foxy R1 fraction collector (ISCO), coupled with UA-6 absorbance detector equipped with chart recorder (ISCO) (sensitivity 0.5, chart speed 150 cm/h). In parallel, the profiles were recorded with TracerDAQ Pro data acquisition software (MCC).

For the mouse liver samples, the same protocol was applied with different lysis conditions. Briefly, dissected tissue pieces were immediately snap-frozen and 100–150 mg was powdered under liquid nitrogen and resuspended in 1 mL tissue polysome lysis buffer (100 mM KCl; 50 mM TRIS-HCl pH = 7.4; 1.5 mM MgCl_2_; 1 mM DTT; 1 mg/mL heparin; 1% Triton X-100; 0.5% sodium deoxycholate and 100 µg/mL CHx (Sigma-Aldrich), supplemented with EDTA-free protease inhibitor cocktail (Roche) and 100 U/mL SuperaseIn RNase inhibitor (Thermo Fisher Scientific)) for 20 min on ice followed by two rounds of centrifugation (15,228× *g*, 20 min, 4 °C). In total, 200 µg RNA were loaded on the gradients and proceeded with the above-mentioned protocol.

Polysomal fractions containing mRNAs associated with three or more ribosomes were precipitated overnight in 100% ethanol, followed by a standard RNA extraction described above.

#### 2.7.2. RNA Sequencing

RNA libraries were prepared using the TruSeq HT Stranded mRNA Kit (Illumina, San Diego, CA, USA). The samples were then run on an Illumina HiSeq 4000 sequencer. The reads were mapped with the STAR v.2.7.0 software to the Ensembl GRCh38 Homo sapiens reference. Biological quality control and summarization were carried out with the PicardTools v.1.141. The table of counts with the number of reads mapping to each gene feature of Ensembl GRCh38 human reference was prepared with HTSeq v0.9.1 (htseq-count). The differential expression analysis was performed with the statistical analysis R/Bioconductor package edgeR v. 3.26.8.

### 2.8. Bioinformatic Analyses

Bioinformatic tools used to analyze our translatomic analysis are described in Appendix A. Data used for gene expression analysis, correlation analysis and survival curves were obtained from Gene Expression Omnibus and The Cancer Genome Atlas (TCGA). For transcriptomic datasets, a deregulation expression pattern of minimum 50% (0.666 < FC < 1.5) was considered to avoid small and biologically irrelevant alterations, as previously described [4,23,24,25].

All the analyses were performed using publicly available software as described in Appendix A.

### 2.9. Statistics

Data were shown as mean ± standard deviation. The two-sided t-test was used for comparison of two groups; one-way ANOVA was used for comparison of three or more groups. Survival data were analyzed using log-rank (Mantel–Cox) tests. Tests used are specified in figure legends. Each sample was measured once. Data were reported as mean ± standard deviation (SD). Statistical tests are described in Appendix A. *p*-values were represented on graphs as follows: * *p* < 0.05, ** *p* < 0.01, *** *p* < 0.001.

## 3. Results

### 3.1. TIA1 Promotes Proliferation and Migration of Hepatic Cancer Cells and Is a Relevant Component of Sorafenib-Induced Stress Granules

Several AUBPs are deregulated in human hepatic cancers (Appendix A) for information about GEO datasets see Appendix A). Among them, the mRNA of TIA1 is strongly upregulated in HCC, as compared to non-tumoral tissues in the TCGA cohort (Figure 1A), a characteristic that was further confirmed in HCC patients described in other transcriptomic datasets (Appendix A). As formerly described, high TIA1 mRNA levels correlate significantly with a poor clinical outcome in HCC patients (Figure 1B, Appendix A) [26].

To investigate the functional relevance of TIA1 in HCC, the proliferation of transformed hepatic cancer cells, which express very high levels of TIA1 as compared to primary hepatocytes (Appendix A), was investigated following TIA1 downregulation by specific siRNA (the efficiency of TIA1 silencing was evaluated by Western blot: see validation blot, Appendix A). As shown in Figure 1C and Appendix A, TIA1 downregulation decreased proliferation of HepG2 and Hepa1-6. No significant alteration of cell cycle distribution nor apoptosis induction (Appendix A) was observed in these cells suggesting a global slowdown of cell cycle in the absence of TIA1. Trans-well migration assays further indicated decreased migratory and invasive capacities of TIA1-deficient cells (Figure 1D), thus supporting an oncogenic activity of TIA1 in transformed hepatic cancer cells as previously suggested [9].

TIA1’s ability to promote SG assembly, which renders cancer cells more resistant to harmful conditions and anticancer treatments, has been reported in several non-liver cancers [11]. Consistent with this, resistance of hepatic cancer cells to sorafenib was reported to be associated with SG formation [16] and TIA1 mRNA expression is increased in sorafenib non-responders (Figure 1E). We observed that TIA1 localizes in SGs stained with the specific marker G3BP1 [27] in Huh7 cells treated with sorafenib (Figure 1F, Appendix A). However, although TIA1 silencing in these cells may impair SG formation (absence of G3BP1 positive SG when TIA1 is silenced in some cells) (Figure 1G), it only slightly impairs sorafenib resistance of hepatic cancer cells (Figure 1H). Of note, sorafenib does not induce SG formation in mouse primary hepatocytes (Appendix A). These data indicate that although TIA1 is incorporated in forming hepatic SGs in hepatic cancer cells, it is not absolutely required for their assembly, nor for the resistance to sorafenib action.

### 3.2. TIA1 Loss Alters the Translation of HCC-Related Genes

TIA1 is a well-known inhibitor of mRNA translation [13,28,29]. To identify relevant targets of TIA1 in HCC, we performed a polysome profiling followed by an RNA sequencing of the highly translated (polysomal) mRNAs of HepG2 cells ± TIA1 (Figure 2A,B, Appendix A). The deregulated genes (FC > =1.5, *p*-value < 0.01, DEGs) appear to be involved in different processes, such as transforming growth factor beta (TGFβ) and protein kinase B (PKB) signaling, cell–cell adhesion or liver development (Figure 2C). The cross-comparison of DEGs in polysome fractions (FC > =1.5, *p*-value < 0.01) with a list of genes deregulated (up- or down-) in HCC retrieved from MetaCore identified 148 candidates genes potentially under the control of TIA1 and involved in HCC (Figure 2D, Appendix A). Among them, 45% classify as ONCs and 37% as TSs according to the CancerMine database (Appendix A). Gene Ontology enrichment analyses indicate that these candidates are associated with various biological processes (e.g., cell proliferation) or pathways (e.g., p53 signaling) associated with carcinogenesis (Appendix A). Among these DEGs (Figure 2E,F), several have an established role in cancerous hallmarks and form a concise network through protein–protein interactions or co-expression patterns (Appendix A), thus suggesting a functional link in HCC development. Moreover, most of them contain an ARE site (Appendix A) and thus may represent direct TIA1 targets. Several ONCs and TSs were respectively down- and upregulated following TIA1 loss (Figure 2E), consistent with the phenotype observed in HepG2 cells (Figure 1). However, TIA1 silencing also led to upregulation of other ONC while reducing the expression of specific TSs (Figure 2F), thus suggesting a pleiotropic function of TIA1 in HCC. A similar trend, but to a lesser extent, was observed for some candidates (i.e., S100A11, BIRC7, ARID2) in Huh7 cells (Appendix A). Finally, among the DEGs, some are also key regulators of hepatic lipid metabolism (e.g., PIK3R1, PPARG, S100A11, Figure 2E,F), suggesting that alterations in TIA1 expression/activity also contribute to fatty liver disease development.

### 3.3. TIA1 Loss Promotes Hepatic Steatosis and Fibrosis

As previously mentioned, HCC often arises in a metabolic context of steatosis, inflammation and fibrosis [1] and TIA1 activity was tightly correlated with several inflammatory diseases [28,30]. However, paradoxically to the TIA1 mRNA upregulation observed in HCC, TIA1 was reported to be downregulated in obese patients [31] and TIA1KO mice were found to develop neuroinflammation, impaired lipid storage and membrane dynamics [32]. Our bioinformatic analyses of publicly available databases or of mouse hepatic tissues did not allow for the observation of consistent alterations in TIA1 mRNA expression in human liver diseases (Appendix A), neither in mouse models of hepatic steatosis nor inflammation (Appendix A). We found, however, that liver samples from TIA1KO mice display a strong deregulated expression pattern of genes associated with the lipid metabolism and inflammation (Figure 3A, Appendix A), many of them being also altered in NAFLD/NASH patients, in mice fed a methionine/choline-deficient (MCD) diet, and, to a lesser extent, in mice fed a classical high-fat-enriched diet (HFD) (Figure 3B, Appendix A). Consistent with these data, TIA1 silencing in HepG2 cells alters the translation of 46 DEGs related to the lipid/glucose metabolism (Figure 3C,D). Lipid accumulation was observed in TIA1-deficient HCC cells (Figure 3E), associated with the specific induction of PPARG, CPT1A and CD36 mRNA levels (Figure 3F). Based on these in vitro data, we assessed in vivo the impact of TIA1 downregulation in the liver for the development of steatosis, inflammation and fibrosis in mice fed an MCD diet. TIA1 expression was silenced in C57BL/6J mice using hepatotropic adeno-associated virus 8 (AAV8) vectors encoding a shRNA against TIA1 (shTIA1) or the respective control (shCTL), prior to submitting the mice to an MCD diet (Figure 3G,H). Despite no changes in body parameters by TIA1 silencing in mice fed an MCD diet (Appendix A), we observed a strong induction of steatosis and fibrosis, but surprisingly not inflammation, in mice having an efficient knockdown of TIA1 in the liver (Figure 3I). This phenotype was further supported by increased levels of ALAT/ASAT in the blood circulation of these mice, as well as of upregulation of markers for fibrosis in their hepatic tissues (Appendix A). Together, these data indicate that TIA1 deficiency in hepatocytes promotes the development in vivo of liver steatosis and fibrosis.

### 3.4. TIA1 Loss Promotes Hepatic Tumor Burden in Mice

To further explore TIA1′s function in vivo in hepatic tumorigenesis occurring in a context of steatosis/fibrosis, TIA1 was knocked down using AAV8 vectors in hepatocytes-specific PTEN knockout (LPTENKO) mice, a mouse model sequentially developing hepatic steatosis, inflammation, fibrosis and hepatic tumors around 11–12 months of age [20,33]. AAV8-mediated TIA1 silencing was performed at 8.5 months of age before tumor appearance (Figure 4A,B). Tumor development was then monitored by computed tomography imaging (CT scan) until 11.5 months of age. Despite no differences observed on body and liver weight, a significant increase in hepatic tumor burden was observed, but not for the volume of single tumors (Figure 4C–E, Appendix A), indicating that tumor initiation was fostered by TIA1 silencing, but not tumor progression and growth. Serum levels of ALAT/ASAT were unchanged in control versus TIA1 knockdown mice at this stage of the disease (Appendix A), but histopathological analyses of tumoral nodules indicated an increased steatotic nodule occurrence in TIA1-deficient livers (Figure 4F). Finally, analysis of polysomal RNAs from livers of CTL and TIA1 knockdown LPTENKO mice (Figure 4G) indicated that consistent with a lack of effect of TIA1 downregulation on tumor growth and progression, markers for proliferation (*Mki67, Pcna, Cdkn1a, Cdkn1b, Cdc25a*) and inflammation (*Il1b, Tgfb*) were unchanged. However, these analyses also revealed that the enhanced tumor initiation triggered by TIA1 downregulation was associated with an increased expression of ONCs previously identified in HepG2 cells (e.g., Spp1, Ccn2) (Figure 4J), following TIA1 silencing. Together, these results show that downregulation of TIA1 protein expression in hepatocytes fosters tumor initiation in a context of fatty liver disease.

### 3.5. TIA1 Protein Expression Is Downregulated in Human Hepatocellular Carcinoma

As shown in Figure 1, upregulation of TIA1 mRNA expression is frequently observed in human HCC but whether this translates in high protein expression was not investigated. Based on the reported tumor-suppressive role for TIA1 in non-liver cancers (Appendix A) and our mouse data showing that TIA1 downregulation fosters tumor initiation (Figure 4), it is uncertain whether TIA1 protein is also upregulated in human HCC in line with its mRNA. To clarify this issue, we performed immunohistochemistry to evaluate TIA1 protein expression in human tissue microarrays (TMA) of HCC patients (antibody validation: Appendix A; Figure 5A–C, separate analyses see: Appendix A).

Importantly, a strong downregulation of TIA1 protein expression was observed in HCC from 56% of the patients (Figure 5B), while TIA1 mRNA downregulation was observed in only 6.36% of the patients (Figure 1C). TIA1 protein downregulation was not associated with the tumoral grade or stage, showing that TIA1 expression is not consistent with the differentiation status (Figure 5C, Appendix A). A similar downregulation of TIA1 protein expression was further observed by proteomic analysis of HCC tissues from patients (Figure 5D). These data, together with the increased hepatic tumor burden in mice following TIA1 knockdown (Figure 4), suggest that TIA1 should not be considered as only an oncogene/tumor promoter. Corroborating this conclusion, an RNA sequencing of ribosome profiling of HCC patients indicates a decrease in TIA1 translation in hepatic tumors as compared to non-tumoral tissues (Figure 5E). Several well-characterized ONCs (e.g., *CCNB1*), also identified in HepG2 cells following TIA1 silencing (Figure 2F), are also upregulated in tumors or correlate with a poor prognosis (Figure 5G,H). These findings further suggest that TIA1 may share similar targets in patients, HCC cell lines and mouse models.

## 4. Discussion

In this study, we found that TIA1 refrains from steatosis/fibrosis development and behaves as a potent hepatic TS, whose expression is lost in human HCC. Furthermore, we uncovered a role for TIA1 in cancer cell proliferation, migration and invasion in hepatic cancer cells. Our study identified a whole network of genes involved in NAFLD/NASH/HCC and under the control of TIA1. Finally, our study indicates that TIA1 colocalizes with SGs, but is not strictly required for their formation in hepatic cancer cells.

Our data revealed discrepant levels of TIA1 mRNA and protein expression in patients with HCC. Indeed, while the TIA1 mRNA expression is frequently found to be upregulated, on the contrary, the TIA1 protein expression is strongly downregulated, suggesting the existence of regulatory feedback loops attempting to restore TIA1 protein expression and activity. Unfortunately, we could not evaluate both TIA1 mRNA and protein level in liver tissues from the same patients as our study exploited publicly available databases (GEO database) and previously published studies using exclusively transcriptomic or proteomic analyses. Therefore, additional efforts on a well-characterized cohort of patients are still required to fully characterize TIA1 regulation in HCC. Nevertheless, our data agree with a decreased TIA1 translation observed in HCC patients (Figure 5D,E) and other studies suggesting a tumor-suppressive function of TIA1 and a downregulation of its expression at the protein level in tumors, while the mRNA level remains unchanged (Appendix A) [14]. Further supporting a regulatory feedback loop for TIA1 protein expression, TIA1 deletion in mice is associated with an upregulation of TIA1 mRNA encoding exon 11, but a decreased expression of the mRNA encoding exon 4 (Appendix A) [32]. The regulatory mechanisms governing TIA1 expression and activity are complex and occur mostly at a post-transcriptional/translational level as previously suggested in other cancers [14,36]. Our analyses of TIA1 mutations and methylation status in HCC patients, as well as of the expression of predicted or validated TIA1 transcription factors, did not reveal drastic changes in HCC (Appendix A). However, a whole panel of microRNAs potentially targeting TIA1 is strongly upregulated in HCC (e.g., mir-19a-3p or miR-487a-3p, etc.) and may significantly block TIA1 mRNA translation (Appendix A) [36,37]. Among them, we found that miR-487a-3p promoted TIA1 downregulation in HCC cells (Appendix A). It is likely that the loss of TIA1 protein expression in tumors results from multiple mechanisms, including targeting by several miRNAs or other non-coding RNAs, but further studies are now required to delineate these mechanisms in detail. Of note, while survival curves correlating high TIA1 mRNA expression with bad prognosis remain meaningful, their interpretation regarding TIA1 functions in hepatocarcinogenesis must be cautious since TIA1 mRNA upregulation appears to reflect a drastic decrease in the protein expression/activity. In this regard, our data support a tumor-suppressive role for TIA1 in HCC initiation, but likely a tumor-promoting function in cancer cells after transformation. Based on these major differences between our in vitro and in vivo models, we suggest a dual and complex role for TIA1 depending on the cellular context, metabolic status and tumor microenvironment.

Because TIA1 is an important regulator of translation [19,29], we privileged a translatome-based approach to uncover its potential targets in HCC cells. Deregulation of the expression of both ONC and TS was associated with TIA1 downregulation, thus making it difficult to predict the role for TIA1 in hepatic carcinogenesis but at the same time highlighting the complex functions it likely has in tumorigenesis (Figure 2). More so, a tumor-suppressive function of TIA1 was observed in vivo for tumor initiation, but not tumor progression or malignancy in LPTENKO mice (Figure 4), while a tumor-promoting role for TIA1 (e.g., on cell proliferation and migration/invasion) was deduced from our in vitro approaches using already transformed hepatic cancer cells (HepG2 and Hepa1-6, Figure 1C and Appendix A). Driving conclusions about the role for specific factors in human carcinogenesis using animal models is already challenging since none of the HCC mouse models currently used in hepatology research faithfully recapitulate all aspects of the human pathology [38]. However, we should be even more cautious when using in vitro cell culture of transformed hepatic cancer, even if the latter are derived from human pathological tissues. Indeed, these cells, which are cultured in the absence of a relevant microenvironment (extracellular matrix, vascularization, immune cells, etc.) and have also undergone major biological changes (derivation, metabolic switch, loss of cell polarity and cell–cell interactions, etc.), have major limitations for the study of carcinogenesis that need to be considered when interpreting experimental results [39]. Moreover, these models do not recapitulate the complex interplay between the different hepatic cell types (stellate cells, Kupffer cells, hepatocytes), which importantly contribute to the development of HCC. In agreement, our data indicate that TIA1 loss is associated with an increase in several secreted factors (e.g., S100 family members, osteopontin), which could potentially foster HCC development by promoting inflammation, hepatic fibrosis, and immune escape of cancer cells, as previously demonstrated [4]. Further illustrating discrepant data arising from in vivo versus in vitro experimental settings, we previously reported that another AUBP tristetraprolin (TTP), or miRNAs such as miR-21, also display opposite functions in vitro versus in vivo in cancer-related processes [10,40].

Our translatomic analyses show an upregulation of lipid metabolism-related transcripts with TIA1 downregulation and an increased accumulation of lipid droplets, concomitant with upregulation of *PPARG*, *CD36* and *CPT1A* expression in Huh7 and HepG2 silenced for TIA1 (Figure 3), suggesting that TIA1 might also represent a key factor in hepatic lipid/glucose metabolism. Our in vivo data with mice fed an MCD diet, which induces severe steatosis and fibrosis [41] further confirmed that TIA1 downregulation fosters lipid accumulation and fibrosis development in the liver (Figure 3). Finally, TIA1 is a well-known attenuator of inflammation, mainly by targeting COX-2 and TNFα [13,28] but we could not detect changes in inflammatory markers between shCTL and shTIA1 mice fed an MCD diet. One possible explanation is that the effects of the MCD diet on inflammation overcome the fine regulation of this specific process by TIA1; however, further studies are required to evaluate the precise role for TIA1 in hepatic inflammation.

An oncogenic or tumor suppressive function of TIA1 could be highly dependent on its subcellular localization as previously suggested in esophageal carcinoma, where TIA1-dependent deregulations of both ONC and TS were found [15] (Appendix A), similarly to our observations. In this regard, evidence indicates that TIA1 in the cytoplasm is mainly associated with SGs, where it regulates translation; while in the nucleus, it mostly governs alternative splicing of specific target genes (e.g., *FAS*, *COL2A1*) [42,43,44]. Whether TIA1 exerts such functions in hepatocytes is currently unknown but is consistent with the variety of ONC (e.g., *ACTG1*, *CHD1L*, *MRPS23*) and TS (e.g., *CD63*) transcripts, whose splicing was modulated by TIA1 silencing in our translatomic analyses (Appendix A). Of note, many of the potential TIA1 targets (Figure 2E,F) are also alternatively spliced and/or known regulators of tumorigenesis (e.g., *CCNB1*, *S100A11*, *SPP1*) [4,45], yet due to their interactions and overlapping functions (Appendix A), it is impossible to identify one target responsible for all the phenotypes observed. Our in vitro findings are, however, in agreement with a previous study suggesting an oncogenic effect of TIA1 in HCC cells by inhibiting the expression of the TS, IGFBP3 [9]. Interestingly, this candidate was not changed in our translatomic analysis nor in TIA1KO mice [32], whose liver transcriptome at 6 months also contains many other ONC and TS (Appendix A).

The pleiotropic role for TIA1 in metabolism and cancer likely also depends on its functional interactions with various other cellular factors differentially expressed in different stages of these hepatic diseases, such as non-coding RNAs or other AUBPs (Appendix A). AUBPs such as HuR and TTP are found respectively overexpressed and downregulated in HCC correlating with the worsening of patients’ survival [40] (Appendix A). However, both TTP and HuR promote steatosis and/or fibrosis development in the liver [40,46]. In addition, AUBPs can also share common cellular targets (Appendix A), suggesting that deregulation, for example of ONC/TS in HCC, likely results from the concerted action of several AUBPs, most of them being altered in HCC (Figure 1). The activity of direct cofactors of TIA1 can also be deeply involved in HCC development as recently reported for TIAR, which in line with our data, has a tumor-suppressive role in HCC [47]. Finally, TIA1′s interaction with lncRNAs, e.g., *MALAT1* and *NEAT1*, which display oncogenic features in HCC (Appendix A), likely also contributes to the tumor-suppressive effect of TIA1, since we could show that TIA1 silencing promotes *NEAT1* upregulation in hepatic cancer cells (Appendix A). 

We finally also demonstrated here that TIA1 is recruited in SGs of hepatocytes as it was previously shown in other cell types [48]. Induction of SG assembly by anti-HCC drugs such as sorafenib has previously been suggested to mediate resistance to these drugs [16]. Here we provide evidence that although TIA1 is required for fully efficient SG assembly induced by sorafenib, its downregulation cannot entirely prevent it, nor fully restore sorafenib sensitivity in hepatic cancer cells. It is currently unclear whether SG assembly in the absence of TIA1 is due to overexpression of other SG components but several (e.g., G3BP1 or UBAP2L) are strongly upregulated in HCC from patients resistant to sorafenib and associated with a poor clinical outcome (Appendix A). Furthermore, whether sorafenib resistance is associated with a specific component of SGs, or whether it requires the assembly of functional SGs, remains to be uncovered. The functional role for SGs and their various components on hepatic cell transformation, proliferation and migration/invasion are still poorly known. A deeper understanding of SGs’ functions in cancer is likely to provide not only relevant biomarkers but also new potential therapeutic targets to fight these diseases, thus encouraging further investigations in this field of research.

## 5. Conclusions

Collectively, our data on TIA1 reinforce the idea that alterations in post-transcriptional regulators of gene expression, such as AUBPs, are relevant in liver diseases and HCC. Currently, very few therapeutic options are available for HCC but also other hepatic cancers (i.e., ICC, hepatoblastoma), and thus a better understanding of TIA1 but also of other AUBPs may provide new biomarkers and/or novel and efficient therapeutic options. However, our study breaks the current view of TIA1 as a strict tumor promoter in HCC and reveals a more complex and context-dependent function. Therefore, an in-depth analysis of TIA1′s function and of its targets is required prior to any therapeutic intervention.

## Figures and Tables

**Figure 1 cancers-14-01704-f001:**
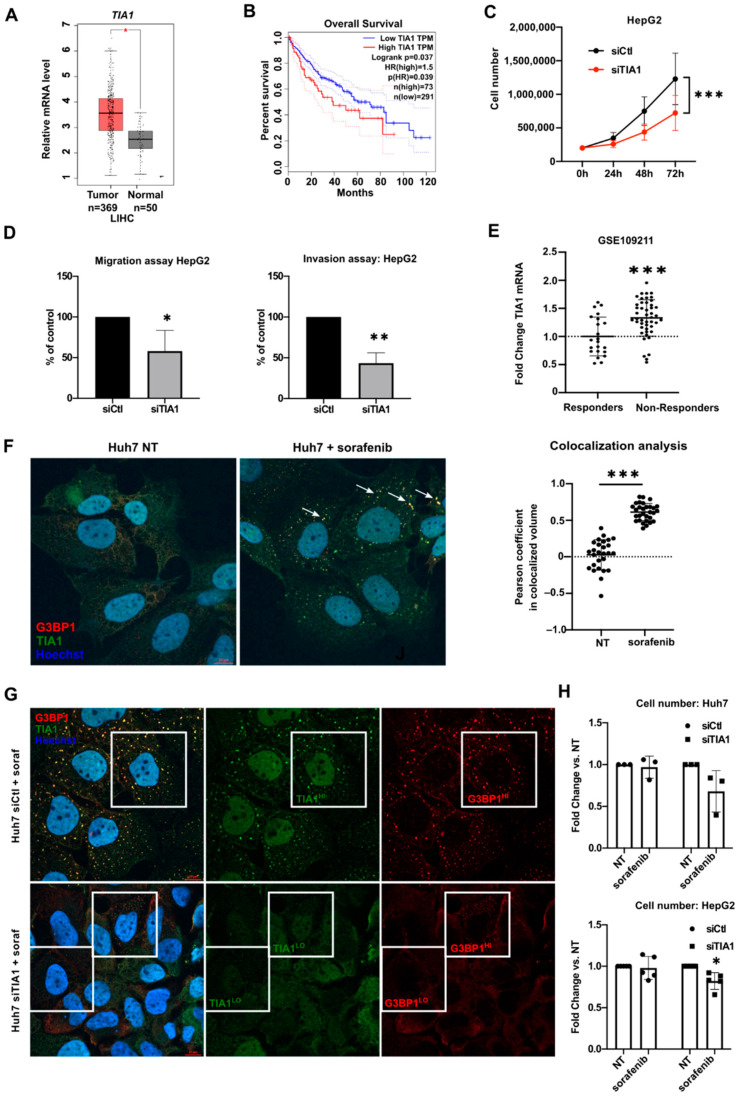
TIA1 deregulation in HCC and impact of TIA1 inhibition in vitro. (**A**) TIA1 expression in HCC tumors vs. healthy tissues (LIHC: TCGA cohort: GEPIA database retrieved 1 June 2021). (**B**) Overall survival of patients based on TIA1 expression in the LIHC cohort (GEPIA2, *p*-value calculated using the Log-rank test). (**C**) Cell number of HepG2 (n = 5) after TIA1 silencing (siTIA1) vs. control cells (siCtl). One-way ANOVA was used. (**D**) Percentage of HepG2 ± siTIA1 (n = 3) migrating through the Boyden chamber, not coated (migration, left) or coated (invasion, right) with Matrigel. (**E**) Relative mRNA expression levels of TIA1 in HCC patients responding to sorafenib vs. non-responders. (**F**) Representative pictures and quantification of TIA1 (green) and G3BP1 (red) signal colocalization in Huh7 ± sorafenib (n = 3). Arrows indicate stress granules. (**G**) Representative pictures of TIA1 (green) and G3BP1 (red) signal in Huh7 ± siTIA1 +. (**H**) Cell number of Huh7 (n = 3) or HepG2 (n = 5) after TIA1 silencing (siTIA1) vs. control cells (siCtl) and sorafenib treatment (24 h of transfection followed by 24 h of treatment with 20 μM of sorafenib). * *p* < 0.05, ** *p* < 0.01, *** *p* < 0.001.

**Figure 2 cancers-14-01704-f002:**
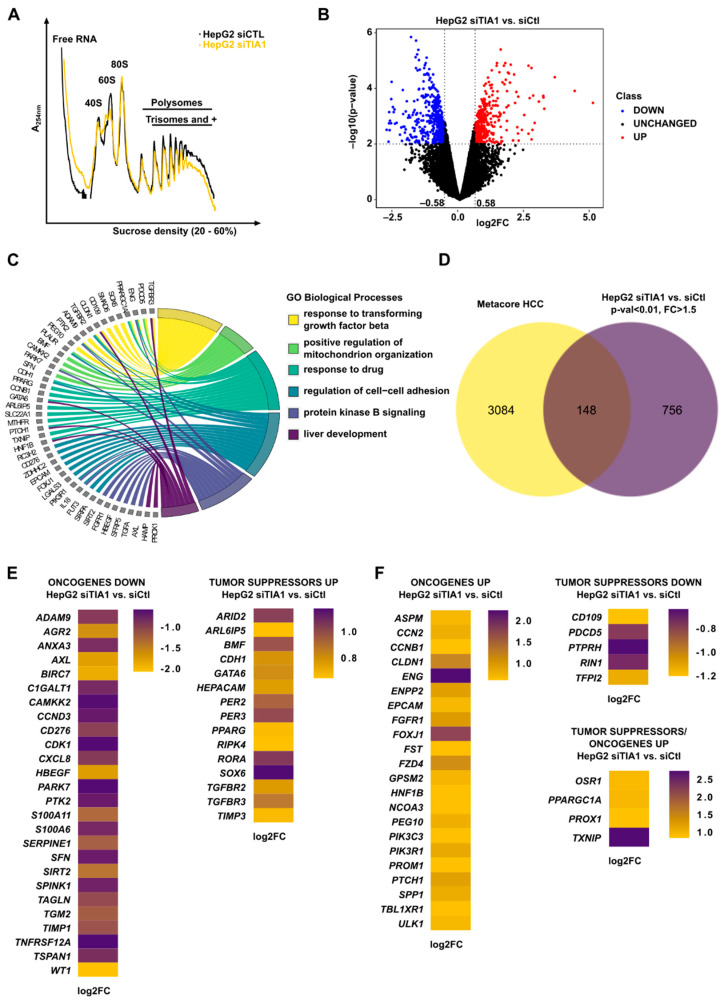
TIA1 regulates the expression of numerous oncogenes and tumor suppressors. (**A**) Representative polysome profiles of HepG2 ± siTIA1. (**B**) Deregulated genes following an RNA sequencing of polysomal (3+) fractions of HepG2 ± siTIA1. (**C**) Most relevant enriched biological processes (BPs) with their genes involved (GO BP, FDR < 0.1). (**D**) Venn diagram showing deregulated genes crossed with a list of genes associated with HCC based on the literature (retrieved from the MetaCore database). (**E**,**F**) The expression of deregulated tumor suppressors (TS) and oncogenes (ONC) in HepG2 ± siTIA1. Data were represented as mean ± SD. Unpaired t-test was used for comparison of two groups, unless specified otherwise.

**Figure 3 cancers-14-01704-f003:**
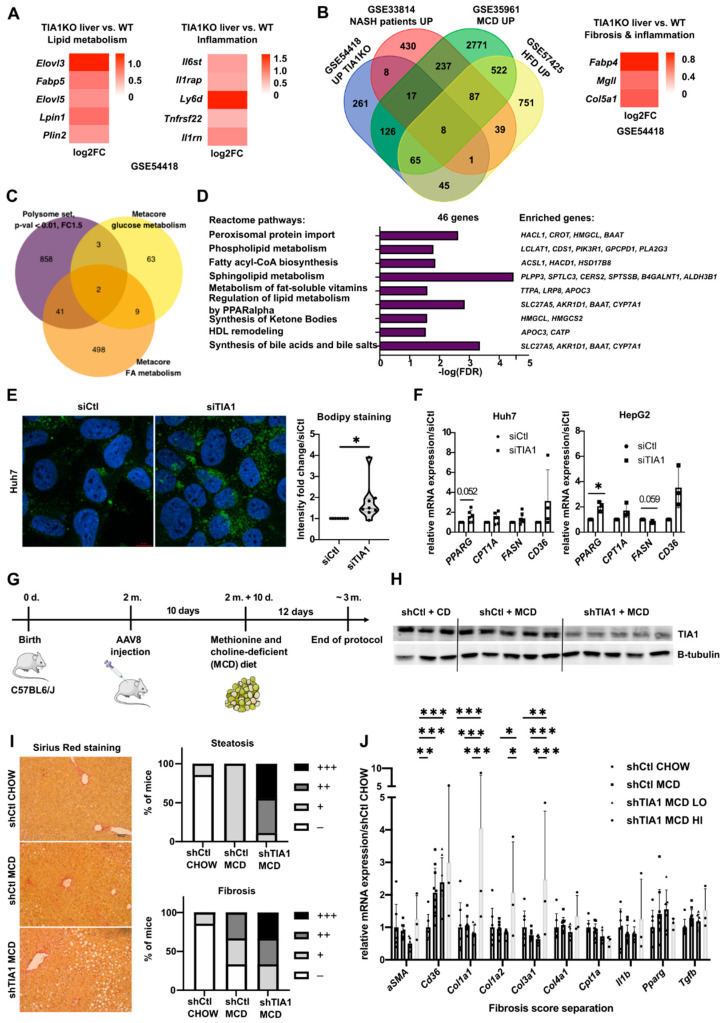
TIA1 prevents steatosis and fibrosis development in mice fed an MCD diet. (**A**) Expression of lipid metabolism and inflammation-associated genes in livers of TIA1KO mice (GSE54518, retrieved 1 October 2021). (**B**) Upregulated genes of livers in TIA1KO mice crossed with genes deregulated in datasets of livers in human NASH patients and mice fed an MCD diet or an HFD. (**C**) Deregulated genes from the HepG2 ± siTIA1 translatomics cross-compared with fatty acid (FA) and glucose metabolism-associated genes. (**D**) GO enrichment analysis of the 46 genes (PANTHER GO-Slim BP, FDR < 0.05). (**E**) Representative pictures and quantifications of BODIPY staining of Huh7 ± siTIA1 (n = 8: 48 h post-transfection). (**F**) Expression of genes related to fatty acid metabolism in Huh7 and HepG2 ± siTIA1 (n = 3–6: 48 h post-transfection). (**G**) Protocol of TIA1 silencing in mouse fed an MCD diet. (**H**) Western blot showing TIA1 silencing in shTIA1/shCtl mice ± MCD diet for 12 days, normalized by β-tubulin. Original blots see Appendix A. (**I**) Representative Sirius red stainings of livers of shTIA1/shCtl mice ± MCD diet, together with the quantifications. (**J**) Relative mRNA expression analysis in shTIA1/shCtl mice ± MCD diet. shTIA1 mice were divided based on their fibrosis score. Data were represented as mean ± SD. Unpaired t-test was used for comparison of two groups and one-way ANOVA with multiple comparisons was used for comparison of three or more groups, unless specified otherwise. * *p* < 0.05, ** *p* < 0.01 and *** *p* < 0.001.

**Figure 4 cancers-14-01704-f004:**
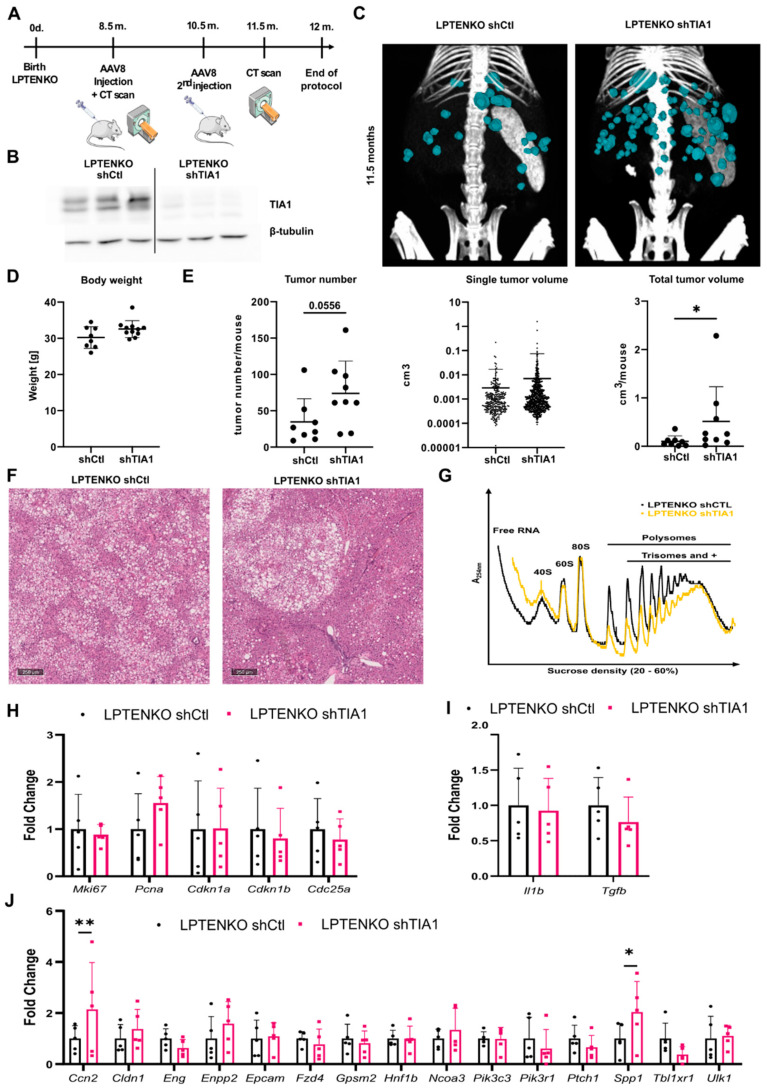
TIA1 loss promotes hepatic tumor burden in vivo. (**A**) Protocol of TIA1 silencing in LPTENKO mice. (**B**) TIA1 protein expression in livers of 11.5 m old LPTENKO mice injected with AAV8 virus against TIA1 and a scramble control (shCtl), normalized by β-tubulin. Original blots see Appendix A. (**C**) Representative 3D reconstructions of computed tomography scans of LPTENKO livers ± shTIA1 at 11.5 months. (**D**) Body weight of LPTENKO mice ± shTIA1 (n = 8 and 11, respectively). (**E**) Tumor number and tumoral volumes in livers of LPTENKO mice ± shTIA1 (n = 8 and 9, respectively). (**F**) Representative pictures of livers of LPTENKO mice ± shTIA1 stained with hematoxylin and eosin. (**G**) Representative polysome profiles of livers of LPTENKO mice ± shTIA1. (**H**,**I**) Relative mRNA expression of (**H**) proliferation and (**I**) inflammatory markers for potential TIA1 targets (**J**) in highly translated fractions of livers of LPTENKO mice ± shTIA1 (n = 5). Data were represented as mean ± SD. Unpaired t-test was used for comparison of two groups. * *p* < 0.05, ** *p* < 0.01.

**Figure 5 cancers-14-01704-f005:**
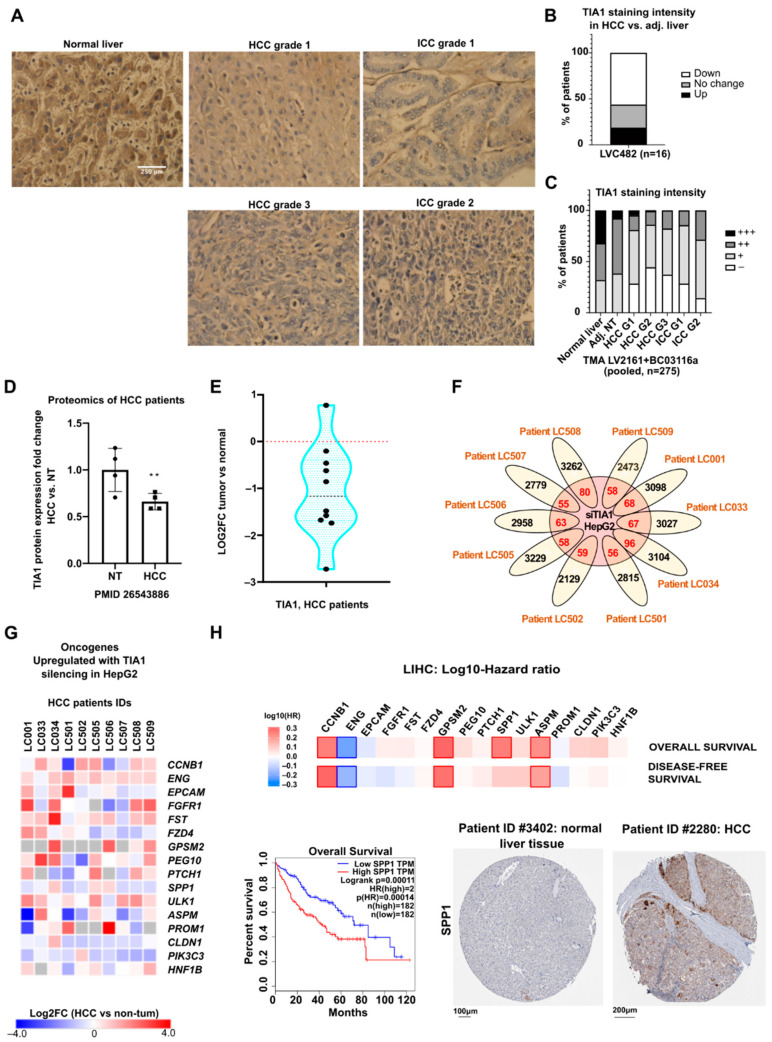
Expression of TIA1 and its targets are deregulated in HCC patients. (**A**) Representative pictures showing TIA1 staining in normal and tumoral liver tissues in a human tissue microarray (TMA). (**B**) Percentage of patients showing a change in TIA1 staining between non-tumoral and HCC tissues in TMA LVC482 (n = 16). (**C**) Intensity of TIA1 staining in human TMAs (n = 275). (**D**) TIA1 protein levels in a proteomic of HCC patients [34]. (**E**) TIA1 mRNA levels in a translatomic of HCC patients [35]. (**F**) Upregulated genes in the translatomic of HCC patients [35] (FC >= 1.5, tumor vs non-tumoral tissues) crossed with genes upregulated in HepG2 siTIA1 translatomics (FC >= 1.5, *p* < 0.01). (**G**) The expression of potential TIA1 targets in the translatomics of HCC patients. (**H**) Hazard ratio of overall and disease-free survival of patients based on mRNA levels of potential TIA1 targets (LIHC, GEPIA), and a survival curve and representative pictures of SPP1 staining in healthy and HCC livers (Human Protein Atlas). Data were represented as mean ± SD. Unpaired t-test was used for comparison of two groups, unless specified otherwise. ** *p* < 0.01.

## Data Availability

The data presented in this study are available in this article (and in the Appendix A) and in YARETA portal. 10.26037/yareta:dqiqskllhzgp5mdkh4b6pt3b7m.

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
