# Peer review of "TIA1 Loss Exacerbates Fatty Liver Disease but Exerts a Dual Role in Hepatocarcinogenesis"

_cancers, 2022, doi:10.3390/cancers14071704_

Round 1

Reviewer 1 Report

In Dolicka et al, the authors discussed the role of TIA1 in hepatocellular carcinoma (HCC). Previously, the mRNA of TIA1 was shown to be higher in HCC in comparison with normal liver tissue. Hence, TIA1 was considered to be an oncogene. In this study, the authors argue that despite of the upregulation of TIA1 mRNA in HCC, the protein levels are not necessarily upregulated. The authors demonstrated that TIA1 loss promotes the expression of oncogenes. They also showed that TIA1 loss promoted loss of tumor suppressors as well. They also showed that loss of TIA1 in mice increased tumor burden. Overall it is interesting that this study challenges the current scientific thought about the role of TIA1 in HCC. However, in its current format, I believe that the manuscript falls short of demonstrating that TIA1 could have a tumor suppression role in HCC. I have the following concerns which the authors should address:

1- The number of samples in the heatmap in Figure 1A and the boxplot in Figure 1B is drastically different. TIA1 expression seems to be heterogeneous even in tumor samples.

2- The break down of FC in Figure 1C is confusing. It is better to check the number of patients in which FC is 1 and above and then show the stats. (FC < 1 and FC > 1).

3- For the Venn diagram in Fig.2D, it is important to categorize the 148 common genes (up-regulated in both or upregulated in siTIA1 only).

4- I don't believe that Figure 4H and Figure 4I are sufficient to demonstrate that there is a tendency of increased proliferation.

5- Figure 5 is very concerning for me. It demonstrates heterogeneity in the protein level of TIA1 in tumor samples. Using data in this figure, the authors argue that mRNA levels of TIA1 are not correlated with TIA1 protein levels. However, Figure 1B clearly shows that tumor samples could have varying levels of mRNA as well. The only way to conclude low correlation between mRNA and protein levels is by detecting the protein and mRNA levels from the same sample.

6- The data in Supplementary Fig. S8A shows a reduction in the protein level of TIA1 following siRNA knockdown. The authors should inform whether this data was used to confirm protein levels of TIA1 for the RNA-Seq experiment in Figure 2.

Author Response

In Dolicka et al, the authors discussed the role of TIA1 in hepatocellular carcinoma (HCC). Previously, the mRNA of TIA1 was shown to be higher in HCC in comparison with normal liver tissue. Hence, TIA1 was considered to be an oncogene. In this study, the authors argue that despite of the upregulation of TIA1 mRNA in HCC, the protein levels are not necessarily upregulated. The authors demonstrated that TIA1 loss promotes the expression of oncogenes. They also showed that TIA1 loss promoted loss of tumor suppressors as well. They also showed that loss of TIA1 in mice increased tumor burden. Overall it is interesting that this study challenges the current scientific thought about the role of TIA1 in HCC. However, in its current format, I believe that the manuscript falls short of demonstrating that TIA1 could have a tumor suppression role in HCC. I have the following concerns, which the authors should address:

We thank the reviewer for their time to review our study and for these positive comments. The tumor suppressive function of TIA1 described in our study is based on an in vivo model (Liver-specific PTEN KO mice) where TIA1 was knockdown with a specific AAV8 encoding an shRNA against TIA1. Although we could not delineate a clear mechanism explaining this tumor suppressive effect, our study challenges the current dogmatic view of TIA1 as a strict tumor-promoting factor in HCC. Our findings are also supported by many other studies reporting tumor suppressive functions of TIA1 but also of its co-factors (TIAR) in non-liver cancers (Supplementary table 5).

Supplementary Table 5. Roles of TIA1 in cancer.

Cancer

RNA level in tumoral tissue

Protein level in tumoral tissue

Function

PMID

colorectal

unchanged

down

TS

28257633

esophageal squamous cell carcinoma

-

Pos. staining in 47% patients

ONC

26958940

cervical (HeLa)

TS

19709424

thyroid cancer

down

TS

31555352

gastric cancer

unchanged

down

TS

30144499

cervical (HeLa)

TS

21284605

many cancers

downregulated

21284605

Understanding the precise mechanism associated to the tumor suppressive function of TIA1 is also difficult, since the loss of TIA1 is associated with the upregulation of many oncogenes, tumor suppressors and the phenotype observed is probably a consequence of these concerted alterations, as described before for other RNA-Binding Proteins [1] but also microRNAs (e.g., miR-21) [2]. One potential explanation for the discrepancies between our in vitro and in vivo models, may reside in the ability of TIA1 to control the expression of secreted factors, which can act in a paracrine manner on non-parenchymal cells in the liver, thereby promoting inflammation, stellate cells activation, immune escape. Among them, we identified S100 family members (e.g., S100A3, S100A6 and S100A11) and osteopontin (SPP1) which are upregulated following TIA1 loss in HepG2 cells. In this scenario, it is therefore difficult to recapitulate all TIA1’s functions by using simple in vitro cell lines, without a physiological environment. An additional paragraph discussing these limitations has been added in the discussion (lines 654-660: unmarked document):

“Moreover, these models do not recapitulate the complex interplay between the different hepatic cell types (stellate cells, Kupffer cells, hepatocytes), which importantly contribute to the development of HCC. In agreement, our data indicate that TIA1 loss is associated to an increase of several secreted factors (e.g., S100 family members, osteopontin), which could

potentially foster HCC development by promoting inflammation, hepatic fibrosis, immune escape of cancer cells as previously demonstrated [4].”

We believe that our study, by providing for the first time an in vivo approach to characterize TIA1’s function in FLD and HCC, is novel and challenges the current studies using in vitro models only. Finally, our translatome analysis will help the scientific community to further characterize the role of this protein in HCC and identify relevant targets involved in these diseases.

1- The number of samples in the heatmap in Figure 1A and the boxplot in Figure 1B is drastically different. TIA1 expression seems to be heterogeneous even in tumor samples.
The figure 1A and 1C have been generated with a transcriptomic dataset from the gene expression omnibus (GEO) database (NCBI: GSE60502), while the figure 1B correspond to the analysis of TIA1 mRNA level in another cohort of patients (LIHC, TCGA cohort) retrieved from the Gepia database. In the Figure 1C, the pooled analysis of all these transcriptomic datasets gathers 330 patients and indicate a significant induction of TIA1 mRNA level. These data are therefore comparable with the figure 1B. We apologize for the lack of clarity and we improved the legend of Figure 1 as follow:

“Figure 1. TIA1 deregulation in HCC and impact of TIA1 inhibition in vitro. (A) Log2 mRNA expression of AUBPs in hepatocellular carcinoma (HCC) patients from a transcriptomic dataset (GSE60502: n=18). (B) TIA1 expression in HCC tumors vs. healthy tissues from another HCC cohort (LIHC: TCGA cohort).”

We agree with the reviewer that we do have a lot of heterogeneity between the tumors and also between the transcriptomic datasets. Beside the fact that HCC is among the most heterogenous cancer [3-6], it is also difficult to understand the difference between these patients, due to the lack of clinical, molecular and histological parameters in these cohorts (etiologies, histologies, treatments, mutations etc). Unfortunately, we don’t have our own well-characterized cohort of patients and thus our data rely on these publicly available databases only. We added a small paragraph in the discussion to explain our limitations (lines 600-604, unmarked document).

2- The break down of FC in Figure 1C is confusing. It is better to check the number of patients in which FC is 1 and above and then show the stats. (FC < 1 and FC > 1).
For proteomic/transcriptomic analyses the classical threshold is 0.5849<Log2FC<-0,5849, which corresponds to 0,666<FC<1.5. These thresholds allow focusing on alteration of minimum 50% (up- and downregulated candidates) and thus avoid focusing on small and biologically irrelevant alterations Additional explanations and references supporting our approach have been added in Materials and Methods (lines 368-374 unmarked document):

“2.8. Bioinformatic analyses
Bioinformatic tools used to analyze our translatomic analysis are described in Supplementary Materials & methods. Data used for gene expression analysis, correlation analysis, survival curves were obtained from Gene Expression Omnibus and The Cancer Genome Atlas (TCGA). For transcriptomic datasets, a deregulation expression pattern of minimum 50% (0,666<FC<1.5) was considered to avoid small and biologically irrelevant alteration, as previously described
[7-10].”

3- For the Venn diagram in Fig.2D, it is important to categorize the 148 common genes (up- regulated in both or upregulated in siTIA1 only).
We apologize for the lack of clarity of this analysis. The goal of this Venn diagram is to identify deregulated transcripts (both up- and downregulated) from our RNAseq, associated to HCC. The metacore database was used to retrieve a list of genes associated to HCC (based on literature) and thus we don’t know which genes are up- or downregulated in this list in HCC.

That’s how we obtained a list of 148 candidates, which was then categorized in oncogenes, tumors suppressors, using the CancerMine database (http://bionlp.bcgsc.ca/cancermine) and literature (pubmed) (Figure 2E). The 148 candidates were also classified in the typical cancerous hallmark, based on literature (Figure S3). However, following the comment of the reviewer, we realized that our explanations might be unclear and thus we provided more detailed explanations in Supplementary Materials and Methods and in the legend of figure 2.

Supplementary Materials: paragraph “Bioinformatic analyses”:

“Identification of HCC-associated genes in the translatomic analysis
To identify genes associated to HCC and deregulated in HepG2 following TIA1 silencing, we retrieved a list of HCC-associated genes with the Metacore database (https://portal.genego.com/). HCC-associated genes and deregulated candidates from our translatomic were compared and represented with a venn diagram. 148 candidates were identified.”

Legend Figure 2.
“TIA1 regulates the expression of numerous oncogenes and tumor suppressors. (A) Rep- resentative polysome profiles of HepG2 ± siTIA1. (B) Deregulated genes following an RNA sequencing of polysomal (3+) fractions of HepG2 ± siTIA1. (C) Most relevant enriched biological processes (BPs) with their genes involved (GO BP, FDR<0.1). (D) Venn diagram showing deregulated genes crossed with a list of genes associated to HCC based on literature (retrieved from the metacore database). (E-F) The expression of deregulated tumor suppressors (TS) and oncogenes (ONC) in HepG2 ± siTIA1. Data was represented as mean ± SD. Unpaired t-test was used for comparison of two groups, unless specified otherwise.”

Moreover, we performed an additional enrichment analysis for these 148 common genes with shinyGO database (http://bioinformatics.sdstate.edu/go/). Our data indicate a significant enrichment of biological processes and pathways associated to carcinogenesis (see figure in the letter for the reviewers) and thus agree with the phenotype observed in HepG2 cells following TIA1 silencing (i.e., decreased proliferation, migration and invasion). These data have been incorporated in Figure S3 (new panel C) and in our Results section (Section 3.2, lines 650-652, unmarked document).

4- I don't believe that Figure 4H and Figure 4I are sufficient to demonstrate that there is a tendency of increased proliferation.
The link between TIA1 and cell proliferation was first evidenced by our in vitro data showing a decreased proliferation following TIA1 downregulation. However, we agree with the reviewer that our conclusion on cell proliferation in vivo is not enough supported by our data. We found an increase of tumor burden following TIA1 loss in LPTENKO mice, without any increase of tumors size. These data suggest an increase of tumor initiation, which correlates with an increased steatotic nodule occurrence in TIA1-deficient livers (Fig. 4F) and an induction of the translation of osteopontin (SPP1) and IGFBP8 (CCN2), which are key drivers/promoters of hepatocarcinogenesis [11-15]. We performed additional qPCRs for other cell proliferation markers, and inflammatory-related genes. Our data did not reveal any significant alterations. Nevertheless, we included the expression data of the additional proliferation markers in Figure 4H (Mki67, Cdkn1a, Cdkn1b and Cdc25a). We therefore mitigate our conclusion as suggested by the reviewer (page 14, section 3.4. TIA1 loss promotes hepatic tumor burden in mice: lines 532-540 in unmarked document):

“Finally, analysis of polysomal RNAs from livers of CTL and TIA1 knockdown LPTENKO mice (Fig. 4G) indicated that consistent with a lack of effect of TIA1 downregulation on tumour growth and progression, markers of proliferation (Mki67, Pcna, Cdkn1a, Cdkn1b, Cdc25a) and inflammation (Il1b, Tgfb) were unchanged. However, these analyses also revealed that the enhanced tumour initiation triggerred by TIA1 downregulation was associated to an increased expression of ONCs previously identified in HepG2 cells (e.g., Spp1, Ccn2) (Fig. 4J), following TIA1 silencing. Together, these results show that downregulation of TIA1 protein expression in hepatocytes fosters tumor initiation in a context of fatty liver disease.”

5- Figure 5 is very concerning for me. It demonstrates heterogeneity in the protein level of TIA1 in tumor samples. Using data in this figure, the authors argue that mRNA levels of TIA1 are not correlated with TIA1 protein levels. However, Figure 1B clearly shows that tumor samples could have varying levels of mRNA as well. The only way to conclude low correlation between mRNA and protein levels is by detecting the protein and mRNA levels from the same sample.

We perfectly agree with the reviewer that the best way to support our conclusion is to analyze the mRNA and the protein levels on the same samples. Unfortunately, we don’t have the possibility to assess both mRNA and protein level of TIA1 in all these patients and we don’t have our own HCC cohort. Nevertheless, we would like to stress the fact that TIA1 mRNA level was decreased in only 6.36% (Figure 1C) of the patients, while our IHC indicate a decrease of TIA1 protein expression in 56% of the patients (Figure 5B). These data were also confirmed by a proteomic analysis and a ribosome sequencing analysis of HCC tumors vs non-tumoral tissue (Figure 5 D and 5E). We therefore believe that our data are important and question the relevance of previous findings based on the mRNA only. Our data are also in agreement with the loss of TIA1 expression in many other cancers, as documented in supplementary table 5). However, following the comment of the reviewer, we mitigate our conclusions, and we discuss this heterogeneity and our limitations in the discussion as follow:

Lines 574-588 (unmarked document): Results: “Importantly, a strong downregulation of TIA1 protein expression was observed in HCC from 56% of the patients (Fig. 5B), while TIA1 mRNA downregulation was observed in only 6,36% of the patients (Figure 1C). TIA1 protein downregulation is not associated to the tumoral grade or stage, showing that TIA1 expression is not consistent with the differentiation status (Fig. 5C, Supplementary Fig. 9). A similar downregulation of TIA1 protein expression was further observed by proteomic analysis of HCC tissues from patients (Fig. 5D). These data suggest that upregulation of TIA1 mRNA levels in HCC hu-man cohorts likely does not correlate with its protein expression and might be due to cellular responses to counteract downregulation of the TIA1 protein.”

Lines 600-604 (unmarked document): Discussion: “Unfortunately, we could not evaluate both TIA1 mRNA and protein level in the same patients as our study exploited publicly available databases (GEO database) and previously published studies using exclusively transcriptomic or proteomic analyses. Therefore, additional efforts on a well- characterized cohort of patients are still required to fully characterize TIA1 regulation in HCC. “

6- The data in Supplementary Fig. S8A shows a reduction in the protein level of TIA1 following siRNA knockdown. The authors should inform whether this data was used to confirm protein levels of TIA1 for the RNA-Seq experiment in Figure 2.
Although, the data shown in supplementary figure S8A used the same model, and experimental condition of the figure 2, these data were used to validate the specificity of our antibody for immunocytochemistry prior to the analysis of TIA1 expression in human tissue microarray (Figure 5). The efficiency of TIA1 silencing with the siRNA used in Figure 2 was evaluated by western blot in HepG2 cells and the data is available in figure 1 of supplementary materials and methods (see figure in the letter for the reviewers). For a better clarity, we added this info in the results:

Line 396-401: “To investigate the functional relevance of TIA1 upregulation in HCC, proliferation of transformed hepatic cancer cells highly expressing TIA1 (Supplementary Fig. 1D-F) was investigated following TIA1 downregulation by specific siRNA (the efficiency of TIA1 silencing was evaluated by western blot: see validation blot, Supplementary Materials & methods). Of note, the expression of TIA1 is high in hepatic cell lines as compared to normal primary hepatocytes.”

Of note, TIA1 knockdown was also confirmed in the RNAseq experiment of Figure 2. Indeed, our data indicate that the siRNA decreases TIA1 mRNA translation, as evidenced by the reduction of the amount of TIA1 mRNA in the polysomes fraction. These data are available in Supplementary table 3: TIA1: log2FC= -1,24 (corresponding to a decrease of 57.6 % of TIA1 expression) with a p-value of 0,014).

Reviewer 2 Report

Authors have set out a systematic study to decipher role of TIA1 in FLD and HCC. It is very interesting to note the discrepancy between transcript and protein profile of TIA1 in HCC. However, this also makes the data interpretation complicated. I have following comments:

  1. I believe the logic of knocking down TIA1 in HepG2 cells, came from the fact that TIA1 transcript is upregulated in HCC. However, looking at the whole data i.e. TIA1 protein is downregulated in HCC, more relevant experiment would have been overexpression of TIA1 followed by transcriptional and functional profiling. Nonetheless, out of all the HCC cell lines profiled for TIA1 protein expression by western blotting Huh7 cells exhibit highest TIA1 protein level. It would be worthwhile to verify key findings of TIA1 knockdown in HepG2 cells in Huh7 cells as well. In parallel, it would be interesting to know the influence of TIA1 overexpression in HepG2 cells.
  2. Influence of TIA1 knockdown on cell proliferation of Hepa1-6 cells is shown however a western blot verifying efficient knockdown in the cells is missing. Please include the data. 
  3. Why cell lines are switched from HepG2 to Huh7 for sofarenib resistance experiment ? Kindly add some explanation or it would be better if the experiment can be done in HepG2 as well.
  4. Similar to above, bodipy staining is done in Huh7 (Fig 3E) instead of HepG2.
  5. In vivo data in very interesting and prove TIA1 as a tumour suppressor as downregulation of TIA1 led to worsening of FLD and more tumour formation. However, the conclusion that it plays dual role and is context dependent between tumour initiation and progression is based on your in vitro data. In vivo, tumour progression was similar in WT versus knockdown groups. So the conclusion and discussion should take this into account. 

Author Response

Authors have set out a systematic study to decipher role of TIA1 in FLD and HCC. It is very interesting to note the discrepancy between transcript and protein profile of TIA1 in HCC. However, this also makes the data interpretation complicated. I have following comments:

We thank the reviewer for his/her time to review our study. We hope that our effort to improve our manuscript will satisfy the reviewer.

  1. I believe the logic of knocking down TIA1 in HepG2 cells came from the fact that TIA1 transcript is upregulated in HCC. However, looking at the whole data i.e. TIA1 protein is downregulated in HCC, more relevant experiment would have been overexpression of TIA1 followed by transcriptional and functional profiling.

We performed additional PCRs for TIA1 in normal human primary hepatocytes (HPH) and in immortalized normal hepatocytes (THLE2 cells). Our data indicate that TIA1 level is strongly upregulated in all the cell lines as compared to normal HPH (up to 30X in HepG2 cells and 20X in Huh7 cells). Similar data were obtained when comparing mouse primary hepatocytes and mouse normal and HCC cell lines (i.e., AML12 and Hepa1-6 respectively). These data have been added in Supplementary Figure S1D and indicate that these cell lines are not fully representative of the situation observed in HCC patients. That’s why we believe that overexpressing TIA1 in these models, where the expression is already very high as compared to normal cells, is not the best approach. In contrary, the silencing TIA1 in a human model allow us to recapitulate at least partially the loss of TIA1 observed in HCC patients (Figure 5). Moreover, the silencing of TIA1 was associated to a phenotype with a decreased proliferation, migration, and invasion (Figure 1 E and 1F), while its overexpression in Huh7 cells using a TIA1-EGFP plasmid did not alter cancer cells proliferation, nor sensitivity to drugs like sorafenib or doxorubicin (see data in the letter for the reviewers).

Nonetheless, out of all the HCC cell lines profiled for TIA1 protein expression by western blotting Huh7 cells exhibit highest TIA1 protein level. It would be worthwhile to verify key findings of TIA1 knockdown in HepG2 cells in Huh7 cells as well. In parallel, it would be interesting to know the influence of TIA1 overexpression in HepG2 cells.

We agree with the reviewer that it is important to ensure that alteration observed in HepG2 can be reproduced in other HCC cell lines. That’s why, in figure 1, we assess the sensitivity of both HepG2 and Huh7 to sorafenib treatment, following TIA1 silencing. In figure 3F, we analyzed the expression of PPARG, CPT1A, FASN and CD36 in both Huh7 and HepG2 cells and our data indicate similar alterations. The impact of TIA1 loss on cell proliferation was also observed in Hepa1-6 cells (Figure S1G). Our data are therefore suggesting a tumor-promoting effect of TIA1 silencing, in agreement with previously published studies on HepG2 [16]. Following the comment of the reviewer, we performed additional qPCRs in Huh7 cells for oncogenes and tumor suppressors (i.e., ARID2, CDH1, S100A6, S100A11, BIRC7) strongly deregulated in HepG2 after TIA1 silencing. Although these analyses were done with total mRNAs and not polysomal mRNAs, our data show a similar trend for ARID2, S100A11 and BIRC7 (Data incorporated in Figure S3: panel F and in the letter for the reviewers).

Regarding TIA1 overexpression in our cell models, we don’t believe that overexpressing more TIA1 in these cell models is likely to provide any mechanistic insights to our study. As mentioned before, all our cells display a huge overexpression of TIA1 as compared to normal primary hepatocytes. Moreover, due to poor transfection efficiency with plasmids, it is challenging to overexpress TIA1 in HepG2 cells, using our TIA1-EGFP vector. We therefore overexpressed TIA1 in Huh7 cells, but we did not observe any alteration of cell proliferation, nor sensitivity to anti-cancerous drugs (sorafenib and doxorubicin: see previous figure).

  1. Influence of TIA1 knockdown on cell proliferation of Hepa1-6 cells is shown however a western blot verifying efficient knockdown in the cells is missing. Please include the data. 

We thank the reviewer for this comment. The WB showing the knock down of TIA1 in Hepa1-6 has been added in Figure S1G. Moreover, the knockdown was also observed at the mRNA level by RT-qPCR analysis.

  1. Why cell lines are switched from HepG2 to Huh7 for sofarenib resistance experiment? Kindly add some explanation or it would be better if the experiment can be done in HepG2 as well.

The experiments for sorafenib sensitivity were performed in both Huh7 and HepG2 cells (Figure 1J). However, the staining for Stress Granules was done in Huh7 cells only because HepG2 cells don’t have contact inhibition and are growing in 3D, thus rendering difficult the interpretation of our microscopy pictures. Nevertheless, we would like to share with the reviewer some pictures showing that sorafenib also promotes SG formation in HepG2 cells (see figure in the letter for the reviewers). Our data are also fitting with a previous study documenting the ability of sorafenib to trigger SG assembly in Huh7 and Hep3B cells [17].

  1. Similar to above, bodipy staining is done in Huh7 (Fig 3E) instead of HepG2.

We thank the reviewer for this comment. BODIPY experiments on HepG2 cells is difficult as, these cells are growing in 3D and form dense clusters, which make difficult the interpretation of our data. For this reason, we privileged the Huh7 cells because these cells are growing in 2D and thus allow an accurate evaluation of intracellular lipid accumulation. However, it should be noted that we found similar alterations of genes associated to lipid metabolism in Huh7, HepG2 and Hep1-6 cells (Figure 3F).

  1. In vivo data in very interesting and prove TIA1 as a tumour suppressor as downregulation of TIA1 led to worsening of FLD and more tumour formation. However, the conclusion that it plays dual role and is context dependent between tumour initiation and progression is based on your in vitro data. In vivo, tumour progression was similar in WT versus knockdown groups. So the conclusion and discussion should take this into account. 

Our in vitro findings suggest that TIA1 promotes HCC cells proliferation, migration and invasion; while in vivo we found that the loss of TIA1 increases hepatic tumor number, without altering the volume of individual tumors. This effect in vivo correlates with an increased steatotic nodule occurrence in TIA1-deficient livers (Fig. 4F) and an induction of the translation of osteopontin (SPP1) and IGFBP8 (CCN2), which are key drivers of hepatocarcinogenesis [11-15]. Our data are therefore suggesting that TIA1 exert a pro-tumorigenic function in vitro and a tumor suppressive function in vivo. We agree with the reviewer that our in vivo data does not show any effect on tumor progression.

Lines 635-640 (unmarked document): “More so, a tumor-suppressive function of TIA1 was observed in vivo for tumor initiation, but not tumor progression or malignancy in LPTENKO mice (Fig. 4), while a tumor-promoting role for TIA1 (e.g., on cell proliferation and migration/invasion) was deduced from our in vitro approaches using already transformed hepatic cancer cells”

Therefore, our conclusion on a dual role of TIA1 was based on the discrepancies between our in vitro and in vivo findings and not on a dual role observed in our in vivo model in different context. As requested by the reviewer, we took this into account, as follow:

Lines 628-630 (unmarked document): “Based on these major differences between our in vitro and in vivo models, we suggest a dual and complex role of TIA1 depending on the cellular context, metabolic status, and tumor microenvironment.”

Unfortunately, we don’t have the explanation for this difference but indeed, the current limitations of in vitro and in vivo models to study hepatic carcinogenesis in vitro models are not fully representative of the complexity of the physiological context, which involves the crosstalks between hepatocytes and non-parenchymal cells (hepatic Stellate cells, endothelial cells, macrophages). These limitations are already discussed in the discussion.

Reviewer 3 Report

cancers-1614817

Title: TIA1 loss exacerbates fatty liver disease but exerts a dual role 3 in hepatocarcinogenesis

Corresponding authors: Cyril Sobolewski and more

In general, the data in present studies are good and support the major conclusions of this manuscript. However, following issues need to be considered prior to considering the manuscript of publication.

Specific comments are as follows:

  1. About abbreviations: The use of abbreviations in the abstract is used separately from the text. That is, it is not necessary to specify abbreviations for abbreviations that are not used repeatedly in the abstract. And, after specifying the full name and abbreviation of a specific term from the introduction part, you can use the abbreviation to the conclusion part. Define TIA1 at the Abstract. When writing an abbreviation, the full name should be written first, followed by the abbreviation.
  2. Materials and Methods section - When naming a particular chemical company, you must provide location information such as company name, city and/or state (abbreviation) and country. Once you have named a company with the information, you should only mention a company’s name thereafter. Examples: Charles River Laboratories, Vector Biolabs, ATCC, etc. Sigma and SIGMA should be Sigma-Aldrich. Sigma-aldrich should be Sigma-Aldrich.
  3. English: The manuscript has numerous grammatical and typographical errors that should be corrected. At Page 7: MgCl2 should be MgCl2. At Page 8: ‘Materials & methods’: As written n the manuscript, it should be written as ‘Materials and Methods’.
  4. Lines 90, 99, and 100 at Page 5: As far as I know, in English sentences, Arabic numerals cannot be the first word of a sentence. If you need to use numbers, you can write sentences like the following two examples. 1) Aliquots of 15 μl of FBS…; 2) Fifty μl aliquots of the cell suspension ....
  5. Page 7: Authors used a centrifuge for the collection of proteins. To describe the centrifugal force, authors should use gravity force, not rpm. They should convert the rpm to a proper gravity force.
  6. Page 8: At Statistics, as far as I know, statistically, *p<0.05, **p><0.01, ***p><0.001, 370 ****p><0.05, **p<0.05, **p><0.01, ***p><0.001, 370 ****p><0.01, and ***p<0.05, **p><0.01, ***p><0.001, 370 ****p><0.001 are usually used. The fourth, ****p<0.0001 is not. You should ask statisticians about this. And the order of notation is usually in the order of *p<0.05, **p><0.01, ***p><0.001, 370 ****p><0.05, **p<0.05, **p><0.01, ***p><0.001, 370 ****p><0.01, and ***p<0.05, **p><0.01, ***p><0.001, 370 ****p><0.001, but here it is indicated in the order of ***p<0.05, **p><0.01, ***p><0.001, 370 ****p><0.001, **p<0.05, **p><0.01, ***p><0.001, 370 ****p><0.01, and *p<0.05, **p><0.01, ***p><0.001, 370 ****p><0.05. please fix it
  7. Figure 2 legend: There is no statistical comparison in Figure 2, but Figure 2 legend has a description of statistics, so delete it.
  8. Figure 4 and 5 legends: *p<0.05, **p><0.01, ***p><0.001, 370 ****p><0.05 and **p<0.05, **p><0.01, ***p><0.001, 370 ****p><0.01 are needed here. Therefore, delete ***p<0.05, **p><0.01, ***p><0.001, 370 ****p><0.001.
  9. The authors capitalize the first letter of words, name of diseases, and chemicals unnecessarily, so find all of them and change them to lowercase. Examples: Adenylate-Uridylate-Rich-Element-binding Proteins, Fatty Liver Disease, Methionine, Choline, etc.
  10. Lines 85 & 86: IF SKP2 and CCNA2 are oncogenes, they should be written in italic.
  11. Line 59: The unit ‘ul’ should be ‘μl’. There are several mistakes on this kind of units in the whole manuscript.
  12. Lines 61, 86, and more: ‘48h’ should be ‘48 h’. A space must be inserted before measurement units. another examples: 1μM, 5μg/ml, etc.
  13. Authors are using two different forms for time unit such as ‘h’ and ‘hour’. Authors should stick on one of them. another examples: ‘min’ and ‘minutes’.
  14. Reference section: Author should consult and peruse carefully recent issues of the journal, cancers, for format and style. The first letter of the title must be in upper case, and the rest must be in lower case.
  15. Although the research hypothesis, research method, and experimental result are excellent, the quality of the manuscript is very low. Therefore, it would be good to have the manuscript revised once again by a colleague who has written many scientific papers like this.

Overall, the manuscript can be considered to publication after major revision as indicated above.

Author Response

In general, the data in present studies are good and support the major conclusions of this manuscript. However, following issues need to be considered prior to considering the manuscript of publication.
We thank the reviewer for their time to review our study. We considered all the comments, and we hope that the reviewer will appreciate our efforts to improve our study.

Specific comments are as follows:

  1. About abbreviations: The use of abbreviations in the abstract is used separately from the text.

    That is, it is not s abbreviation of a specific term from the introduction part; you can use the abbreviation to the conclusion part. Define TIA1 at the Abstract. When writing an abbreviation, the full name should be written first, followed by the abbreviation.
    We thank the reviewer for noticing these mistakes that have been corrected now.

  2. Materials and Methods section - When naming a particular chemical company, you must provide location information such as company name, city and/or state (abbreviation) and country. Once you have named a company with the information, you should only mention a company’s name thereafter. Examples: Charles River Laboratories, Vector Biolabs, ATCC, etc. Sigma and SIGMA should be Sigma-Aldrich. Sigma-aldrich should be Sigma-Aldrich. We apologize for the mistakes. We have fixed all these issues in the Materials and Methods.

  3. English: The manuscript has numerous grammatical and typographical errors that should be corrected. At Page 7: MgCl2 should be MgCl2. At Page 8: ‘Materials & methods’: As written n the manuscript, it should be written as ‘Materials and Methods’.
    We have corrected all the grammatical and typographical errors identified by the reviewer. The manuscript has been carefully reviewed by all co-authors.

  4. Lines 90, 99, and 100 at Page 5: As far as I know, in English sentences, Arabic numerals cannot be the first word of a sentence. If you need to use numbers, you can write sentences like the following two examples. 1) Aliquots of 15 μl of FBS...; 2) Fifty μl aliquots of the cell suspension ....

    The errors noticed by the reviewer were all corrected.

  5. Page 7: Authors used a centrifuge for the collection of proteins. To describe the centrifugal force, authors should use gravity force, not rpm. They should convert the rpm to a proper gravity force.
    We have converted the rpm in rcf as suggested by the reviewer.

  6. Page 8: At Statistics, as far as I know, statistically, *p<0.05, **p><0.01, ***p><0.001, 370 ****p><0.05, **p<0.05, **p><0.01, ***p><0.001, 370 ****p><0.01, and ***p<0.05, **p><0.01, ***p><0.001, 370 ****p><0.001 are usually used. The fourth, ****p<0.0001 is

not. You should ask statisticians about this. And the order of notation is usually in the order of *p<0.05, **p><0.01, ***p><0.001, 370 ****p><0.05, **p<0.05, **p><0.01, ***p><0.001, 370 ****p><0.01, and ***p<0.05, **p><0.01, ***p><0.001, 370 ****p><0.001, but here it is indicated in the order of ***p<0.05, **p><0.01, ***p><0.001, 370 ****p><0.001, **p<0.05, **p><0.01, ***p><0.001, 370 ****p><0.01, and *p<0.05, **p><0.01, ***p><0.001, 370 ****p><0.05. please fix it

The fourth, ****p<0.0001 is used by GraphPad Prism software. We asked a statistician and indeed the p<0,0001 (***) is usually not employed and is not really necessary. We therefore removed this threshold in our figures, supplementary figures and corresponding legends. Now, in the statistic section, page 8, p-values are indicated as follow:

P-values were represented on graphs as follows: *p<0.05, **p<0.01, ***p<0.001

  1. Figure 2 legend: There is no statistical comparison in Figure 2, but Figure 2 legend has a description of statistics, so delete it.
    The description of the statistic in the legend of figure 2 has been deleted.

  2. Figure 4 and 5 legends: *p<0.05, **p><0.01, ***p><0.001, 370 ****p><0.05 and **p<0.05, **p><0.01, ***p><0.001, 370 ****p><0.01 are needed here. Therefore, delete ***p<0.05, **p><0.01, ***p><0.001, 370 ****p><0.001.
    The issue with the P-value thresholds was corrected in all figures, supplementary figures and in the corresponding legends, as suggested by the reviewer.

  3. The authors capitalize the first letter of words, name of diseases, and chemicals unnecessarily, so find all of them and change them to lowercase. Examples: Adenylate-Uridylate-Rich- Element-binding Proteins, Fatty Liver Disease, Methionine, Choline, etc.
    We thank the reviewer for noticing these mistakes that have been corrected now throughout the manuscript.

  4. Lines 85 & 86: IF SKP2 and CCNA2 are oncogenes, they should be written in italic.

    We thank the reviewer for noticing these mistakes that have been corrected according to the official nomenclature of gene name.

  5. Line 59: The unit ‘ul’ should be ‘μl’. There are several mistakes on this kind of units in the whole manuscript.
    We have corrected these mistakes throughout the manuscript.

  6. Lines 61, 86, and more: ‘48h’ should be ‘48 h’. A space must be inserted before measurement units. another examples: 1μM, 5μg/ml, etc.
    We have corrected these mistakes throughout the manuscript.

  7. Authors are using two different forms for time unit such as ‘h’ and ‘hour’. Authors should stick on one of them. another examples: ‘min’ and ‘minutes’.
    We have harmonized the units in our manuscript.

  8. Reference section: Author should consult and peruse carefully recent issues of the journal, cancers, for format and style. The first letter of the title must be in upper case, and the rest must be in lower case.
    We followed the instructions for authors of Cancers, as indicated below:

    Your references may be in any style, provided that you use the consistent formatting throughout. It is essential to include author(s) name(s), journal or book title, article or chapter title (where required), year of publication, volume and issue (where appropriate) and pagination. DOI numbers (Digital Object Identifier) are not mandatory but highly

encouraged. The bibliography software package EndNote, Zotero, Mendeley, Reference Manager are recommended.”

Moreover, we used the MDPI endnote file to format our references (https://www.mdpi.com/authors/references). Our format is therefore in agreement with Cancers.

15. Although the research hypothesis, research method, and experimental result are excellent, the quality of the manuscript is very low. Therefore, it would be good to have the manuscript revised once again by a colleague who has written many scientific papers like this.
We thank the reviewer for their positive comments regarding our methodologies and our results. We apologize for all the numerous grammatical/typographical errors in our manuscript. We have corrected all these mistakes and the manuscript has been carefully revised by all the co-authors. We hope that the reviewer will appreciate our efforts to improve the quality of our manuscript.

Overall, the manuscript can be considered to publication after major revision as indicated above.
We thank again the reviewer for their time and pertinent comments, which considerably helped us to improve our study.

Round 2

Reviewer 1 Report

In this round of revision, the authors have further clarified further aspects of the study, methodology and discussed fairly the limitations of their study.

I appreciate that the authors have toned down the conclusion of Figure 4. I still believe that this study is interesting because it challenges the thought about TIA1 being an oncogene. I have the following recommendations before considering the study for publication.

1- I still feel that figure 1A-C is confusing because it is using two different datasets with varying number of libraries. Figure 1A could be moved to supplementary.

2- Minor suggestion for Figure 1B: Add a y-axis title.

3- Another suggestion for Figure 5: Looking at the data in Figure 5D-E, looks like the cohort had lower expression of TIA1 mRNA as well. Hence, it is better not to discuss about correlation even if toned down. I recommend stating that in your cohort of study, TIA1 expression was generally lower and given the data in previous figures, TIA1 should not be considered as an oncogene only. Emphasize about heterogeneity of TIA1 expression in HCC in the discussion.

Author Response

In this round of revision, the authors have further clarified further aspects of the study, methodology and discussed fairly the limitations of their study. I appreciate that the authors have toned down the conclusion of Figure 4. I still believe that this study is interesting because it challenges the thought about TIA1 being an oncogene. I have the following recommendations before considering the study for publication.

We thank the reviewer for their time and the constructive comments, which helped us to improve the quality of our manuscript.

1- I still feel that figure 1A-C is confusing because it is using two different datasets with varying number of libraries. Figure 1A could be moved to supplementary.
We moved Figure 1A and 1C in supplementary figure 1, as suggested by the reviewer.

2- Minor suggestion for Figure 1B: Add a y-axis title.

We added a y-axis title: “Relative mRNA level”.

3- Another suggestion for Figure 5: Looking at the data in Figure 5D-E, looks like the cohort had lower expression of TIA1 mRNA as well. Hence, it is better not to discuss about correlation even if toned down. I recommend stating that in your cohort of study, TIA1 expression was generally lower and given the data in previous figures, TIA1 should not be considered as an oncogene only. Emphasize about heterogeneity of TIA1 expression in HCC in the discussion.

Since TIA1 protein and mRNA levels were not measured in the same cohort of patients, we agree with the reviewer that we should not discuss about “correlation”. We therefore adapted our conclusion following the recommendation of the reviewer as follow:

Line 636-650 (version with trackchange): “Importantly, a strong downregulation of TIA1 protein expression was observed in HCC from 56% of the patients (Fig. 5B), while TIA1 mRNA downregulation was observed in only 6,36% of the patients (Figure 1C). TIA1 protein downregulation was not associated to the tumoral grade or stage, showing that TIA1 expression is not consistent with the differentiation status (Fig. 5C, Supplementary Fig. 9). A similar downregulation of TIA1 protein expression was further observed by proteomic analysis of HCC tissues from patients (Fig. 5D). These data, together with the increased hepatic tumor burden in mice following TIA1 knockdown (Figure 4), suggest that TIA1 should not be considered as an oncogene/tumor promoter only.”

Regarding the heterogeneity of TIA1 expression in HCC, we have already a paragraph in our discussion, where we discussed the limitations of our study:

Line 566-670 (version with trackchange): “Our data revealed discrepant levels of TIA1 mRNA and protein expression in patients with HCC. Indeed, while the TIA1 mRNA expression is frequently found upregulated, the TIA1 protein expression is on the contrary strongly downregulated suggesting the existence of regulatory feedback loops attempting to restore TIA1 protein expression and activity. Unfortunately, we could not evaluate both TIA1 mRNA and protein level in liver tissues of the same patients as our study exploited publicly available databases (GEO database) and previously published studies using exclusively transcriptomic or proteomic analyses. Therefore, additional efforts on a well- characterized cohort of patients are still required to fully characterize TIA1 regulation in

HCC. Nevertheless, our data agree with a decreased TIA1 translation observed in HCC patients (Fig. 5D-E) and other studies suggesting a tumor-suppressive function of TIA1 and a downregulation of its expression at the protein level in tumors, while the mRNA level remains unchanged (Supplementary Table 5)”

Reviewer 2 Report

The authors have addressed the concerns and in their letter have mentioned about incorporating some data in the supplementary figures however, the files I have downloaded do not show updated supplementary figures. Since the data was also include in the letter I am satisfied with the additional data and explanations. So please check that the supplementary figures are all updated before publication.

Author Response

The authors have addressed the concerns and in their letter have mentioned about incorporating some data in the supplementary figures however, the files I have downloaded do not show updated supplementary figures. Since the data was also include in the letter I am satisfied with the additional data and explanations. So please check that the supplementary figures are all updated before publication.

We thank the reviewer for their time and pertinent comments, which considerably helped us to improve our study. Our supplementary files with updated figures were uploaded during our re-submission. The supplementary materials are in a zip file: “Cancers_1614817”, in a word file entitled: “Cancers_Dolicka et al_Supplementary Materials_R2”. We re-uploaded the file for the submission of the revised version.

Reviewer 3 Report

cancers-1614817-v2

Title: TIA1 loss exacerbates fatty liver disease but exerts a dual role 3 in hepatocarcinogenesis

Corresponding authors: Cyril Sobolewski and more.

At the request of reviewers, the authors made appropriate corrections, but there are still some additional corrections to be made. I'm rather sorry that the authors didn't properly make further corrections, especially for the things I pointed out.

Specific comments are as follows:

  1. Materials and Methods section - When naming a particular chemical company, you must provide location information such as company name, city, and/or state (abbreviation for the states of USA) and country. Once you have named a company with the information, you should only mention a company’s name thereafter. The abbreviation of the states and the United States of America (USA) should be properly corrected. Examples: Vector Biolabs, Sekisui Xenotech, Thermo Fisher Scientific, Takara Bio, Zeiss, Molecular Probes, etc.
  2. Line 118: Pa should be PA.
  3. Line 164: Delete US.
  4. Lines 312, 320: US should be USA.
  5. Lines 204: As far as I know, in English sentences, Arabic numerals cannot be the first word of a sentence. Rewrite the sentence.
  6. Lines 341 and 352: Authors used a centrifuge for the collection of proteins. To describe the centrifugal force, authors should use gravity force, not rpm. They should convert the rpm to a proper gravity force.
  7. Line 448: P53 should be p53.
  8. Reference section: Author should consult and peruse carefully recent issues of the journal, cancers, for format and style. The first letter of the title must be in upper case, and the rest must be in lower case. Examples: 1, 2, 6, 19, 25, 30, etc.
  9. Titles of the journals at the Reference section should be corrected. Examples: 14, 48, etc.

Overall, the manuscript can be considered to publication after minor revision as indicated above.

Author Response

At the request of reviewers, the authors made appropriate corrections, but there are still some additional corrections to be made. I'm rather sorry that the authors didn't properly make further corrections, especially for the things I pointed out.
We apologize for omitting some mistakes in our revised manuscript. We carefully checked the issues raised by the reviewer and we fixed them. We thank again the reviewer for his help.

Specific comments are as follows:
Materials and Methods section - When naming a particular chemical company, you must provide location information such as
company name, city, and/or state (abbreviation for the states of USA) and country. Once you have named a company with the information, you should only mention a company’s name thereafter. The abbreviation of the states and the United States of America (USA) should be properly corrected. Examples: Vector Biolabs, Sekisui Xenotech, Thermo Fisher Scientific, Takara Bio, Zeiss, Molecular Probes, etc.
We carefully checked the Materials/Methods section and we fixed all the issues mentioned by the reviewer.

Line 118: Pa should be PA.

We have corrected this mistake

Line 164: Delete US.

We have corrected this mistake

Lines 312, 320: US should be USA.

We have corrected these mistakes.

Lines 204: As far as I know, in English sentences, Arabic numerals cannot be the first word of a sentence. Rewrite the sentence.
We have corrected this mistake.

Lines 341 and 352: Authors used a centrifuge for the collection of proteins. To describe the centrifugal force, authors should use gravity force, not rpm. They should convert the rpm to a proper gravity force. Following the previous comment of the reviewer, all “rpm” were replaced by “rcf” (relative centrifugal force), which corresponds to the gravity force. “RCF” were replaced by “g” for a better clarity.

Line 448: P53 should be p53.

We have corrected this mistake.

Reference section: Author should consult and peruse carefully recent issues of the journal, cancers, for format and style. The first letter of the title must be in upper case, and the rest must be in lower case. Examples: 1, 2, 6, 19, 25, 30, etc.
As explained in our previous answer, we followed the instructions for authors of Cancers, as indicated below:

Your references may be in any style, provided that you use the consistent formatting throughout. It is essential to include author(s) name(s), journal or book title, article or chapter title (where required), year of publication, volume and issue (where appropriate) and pagination. DOI numbers (Digital Object Identifier) are not mandatory but highly encouraged. The bibliography software package EndNote, Zotero, Mendeley, Reference Manager are recommended.”

Nevertheless, we have corrected the titles as suggested by the reviewer.

Titles of the journals at the Reference section should be corrected. Examples: 14, 48, etc.

The titles of the journals have been corrected.

Overall, the manuscript can be considered to publication after minor revision as indicated above.

We thank again the reviewer for their time and constructive comments.
